# Erythrocytes retain hypoxic adenosine response for faster acclimatization upon re-ascent

Anren Song[1], Yujin Zhang[1], Leng Han[1], Gennady G. Yegutkin[2], Hong Liu[1,3], Kaiqi Sun[1,3], Angelo D'Alessandro[4], Jessica Li[1], Harry Karmouty-Quintana[1], Takayuki Iriyama[1,5], Tingting Weng[1], Shushan Zhao[1,6], Wei Wang[1,7], Hongyu Wu[1], Travis Nemkov[8], Andrew W. Subudhi[8], Sonja Jameson-Van Houten[8], Colleen G. Julian[8], Andrew T. Lovering[8], Kirk C. Hansen[4], Hong Zhang[9], Mikhail Bogdanov[1], William Dowhan[1], Jianping Jin[1], Rodney E. Kellems[1,3], Holger K. Eltzschig[10], Michael Blackburn[1,3], Robert C. Roach[8] & Yang Xia[1,3,7]

Faster acclimatization to high altitude upon re-ascent is seen in humans; however, the molecular basis for this enhanced adaptive response is unknown. We report that in healthy lowlanders, plasma adenosine levels are rapidly induced by initial ascent to high altitude and achieved even higher levels upon re-ascent, a feature that is positively associated with quicker acclimatization. Erythrocyte equilibrative nucleoside transporter 1 (eENT1) levels are reduced in humans at high altitude and in mice under hypoxia. eENT1 deletion allows rapid accumulation of plasma adenosine to counteract hypoxic tissue damage in mice. Adenosine signalling via erythrocyte ADORA2B induces PKA phosphorylation, ubiquitination and proteasomal degradation of eENT1. Reduced eENT1 resulting from initial hypoxia is maintained upon re-ascent in humans or re-exposure to hypoxia in mice and accounts for erythrocyte hypoxic memory and faster acclimatization. Our findings suggest that targeting identified purinergic-signalling network would enhance the hypoxia adenosine response to counteract hypoxia-induced maladaptation.

[1] Department of Biochemistry and Molecular Biology, The University of Texas Health Science Center at Houston, Houston, Texas 77030, USA. [2] Medicity Research Laboratory, University of Turku, 20520 Turku, Finland. [3] Graduate School of Biomedical Sciences, University of Texas Health Science Center at Houston, Houston, Texas 77030, USA. [4] Department of Biochemistry and Molecular Genetics, University of Colorado, Aurora, Colorado 80045, USA. [5] Department of Obstetrics and Gynecology, Faculty of Medicine, The University of Tokyo, Tokyo 113-8655, Japan. [6] Department of Orthopedics, Xiangya Hospital, Central South University, Changsha, 410008 Hunan, China. [7] Department of Nephrology, Xiangya Hospital, Central South University, Changsha, 410008 Hunan, China. [8] Altitude Research Center, Department of Emergency Medicine University of Colorado School of Medicine, Aurora, Colorado 80045, USA. [9] Department of Pathology, MD Anderson Cancer Center, Houston, Texas 77030, USA. [10] Organ Protection Program, Department of Anesthesiology, University of Colorado School of Medicine, Aurora, Colorado 80045, USA. Correspondence and requests for materials should be addressed to Y.X. (email: yang.xia@uth.tmc.edu).

nsufficient tissue oxygen ($O_2$) availability (hypoxia) is a common and dangerous physiological and pathological condition. To survive hypoxia, our body undergoes a number of adaptive responses to promote $O_2$ delivery to peripheral tissues to cope with this challenging condition[1,2]. For more than a century, substantial effort has focused on understanding the integrated physiological response to high altitude in normal individuals, including the hypoxia ventilator response, diuresis, increased cardiac output, improved oxygen-carrying capacity, cerebral blood flow and erythropoiesis[3,4]. In this way, our body gradually acclimatizes to high altitude with decreased acute mountain sickness (AMS), improved exercise performance and restored cognitive function[5]. The inability to adjust to high altitude may lead to pulmonary or cerebral oedema, poor cardiovascular function and even death[6–8]. An intriguing and consistent observation is that following descent to lower elevations, humans retain the acclimatization to high altitude and show a much faster acclimatization upon re-ascent for some time[5]. Surprisingly, the improved and faster acclimatization to high altitude upon re-ascent does not correspond to increased arterial oxygenation and increased erythropoiesis ($CaO_2$ is lower), two common physiological responses associated with the initial adaptive response[5], suggesting that other factors are responsible for the facile response upon re-ascent to high altitude.

Like normal individuals facing high-altitude hypoxia, patients with cardiovascular diseases, respiratory diseases, haemolytic disorders and certain cancers are confronted with pathological hypoxia, which participates in disease progression, organ damage and failure[3,6–10]. As with high-altitude hypoxia in normal individuals, these patients are able to trigger an adaptive response to pathological hypoxic conditions to survive. Thus, adaptive responses to hypoxia are common in normal healthy individuals facing high-altitude hypoxia and patients facing pathological hypoxia to counteract tissue hypoxia for survival. It is extremely difficult to dissect out the adaptive response to hypoxia in patients because of complicated factors associated with time course of disease progression, with disease-specific tissue damage and variables including genetic predisposition and environmental factors. Thus, understanding cellular and molecular mechanisms through which altitude acclimatization occurs in normal humans may lead to new insights regarding adaption to hypoxia and identify potential targets to counteract the maladaptive effects of hypoxia.

Extracellular adenosine levels are tightly controlled at multiple steps including its generation from ATP by ectonucleotidases (CD39 and CD73), degradation by adenosine deaminase (ADA) and elimination by equilibrative nucleotide transporters (ENTs). For the past 20 years, substantial studies have focused on extracellular adenosine generation under stress or hypoxic conditions and its function via activation of its specific surface receptors on multiple cell types[11–13]. For example, early studies showed that genetic deletion of CD73 or CD39 abolishes acute extracellular accumulation of adenosine and leads to severe hypoxic tissue damage[14,15]. Once extracellular adenosine is produced, it elicits multiple functions including anti-vascular leakage, anti-inflammation and vasodilation to protect tissue damage under acute hypoxia setting[16–18]. More recent studies have revealed a protective role of extracellular adenosine activating AMP-mediated protein kinase through ADORA2B receptor in the normal erythrocyte to induce 2,3-bisphosphoglycerate (2,3-BPG) production and subsequently promote oxygen delivery to counteract hypoxic tissue damage[19]. In contrast, due to the mutation of β-haemoglobin in sickle cell disease (SCD; HbS), elevated adenosine signalling via ADORA2B-induced production of 2,3-BPG in the SCD erythrocyte becomes detrimental because it triggers deoxygenated HbS, polymerization and eventually sickling,

a central pathophysiology of SCD[20]. Besides SCD, numerous studies showed that sustained accumulated adenosine signalling via ADORA2B receptors contributes to pathophysiology of multiple chronic settings including chronic kidney diseases, pulmonary fibrosis, priapism, preeclampsia and chronic pain[20–25]. However, whether the hypoxia adenosine response is a common and key regulatory mechanism underlying initial acclimatization and subsequent retention during re-ascent remains unclear.

Here by combining human high-altitude studies and mouse genetic studies, we discovered that CD73-depedent elevation of plasma adenosine signalling via ADORA2B-mediated protein kinase A (PKA) phosphorylation, ubiquitination and proteasome degradation of erythrocyte ENT1 is a novel feed-forward signalling network underlying initial hypoxic adaptation and retention upon re-exposure. These findings reveal significant new insight to the molecular basis underlying adaptation to physiological and pathological hypoxia and thereby open up novel therapeutic possibilities for the potential consequences of exposure to hypoxia.

## Results

**Plasma adenosine associates with altitude acclimatization.** To assess the hypoxic adenosine response at high altitude and upon re-ascent, we measured the levels of purinergic components in plasma from human volunteers participating in the AltitudeOmics study[5] (Fig. 1a). Unexpectedly, we found that plasma adenosine levels were significantly higher upon re-ascent to 5,260 m for 1 day, after spending 7 or 21 days at 1,525 m (Post7/21), than on the first hypoxia exposure for 1 day (ALT1; Fig. 1b). Thus, plasma adenosine levels were induced by initial hypoxia in a time-dependent manner and remained at significantly higher levels upon re-ascent compared with the first hypoxic exposure (ALT1).

Soluble CD73 (sCD73) is an important purinergic component of the hypoxic adenosine response in acute hypoxia and functions to generate extracellular adenosine. Therefore, we measured sCD73 activity in normal individuals at sea level (SL), at high altitude and upon re-ascent. Intriguingly, sCD73 activity displayed the same trend as elevated extracellular adenosine, a significant time-dependent elevation at high altitude and a further induction upon re-ascent (Fig. 1c). No significant differences in plasma adenosine and sCD73 activity were seen between males and females upon first ascent to high altitude and following re-ascent (Supplementary Fig. 1a,b). Thus, we provide human *in vivo* evidence that sCD73 activity was induced during initial exposure to high altitude and retained its high activity upon re-ascent.

Our discoveries of increased plasma adenosine and sCD73 activity in response to high altitude and their rapid and even greater elevation on re-ascent raise a possibility that these biomarkers are physiological signals that contribute to rapid physiological acclimatization upon re-ascent. To test this possibility, we evaluated the relationship between the increase in these two biochemical parameters (that is, plasma adenosine level and sCD73 activity) to the improvement of the physiological acclimatization on Post7/21 (day 1 upon re-ascent) relative to ALT1 (day 1 of the initial ascent). Specifically, we performed Pearson Correlation analysis to determine whether the elevation of plasma adenosine and sCD73 on Post7/21 relative to ALT1 were associated with the reduction of AMS-C composite score, which is commonly used to evaluate physiological adaptation to high altitude. We observed a significant correlation between the increase in plasma adenosine levels and the decrease in AMS-C composite score on Post7/21 compared with ALT1 (Pearson correlation $r = -0.64$, $P = 0.003$; Fig. 1d). However, we did not

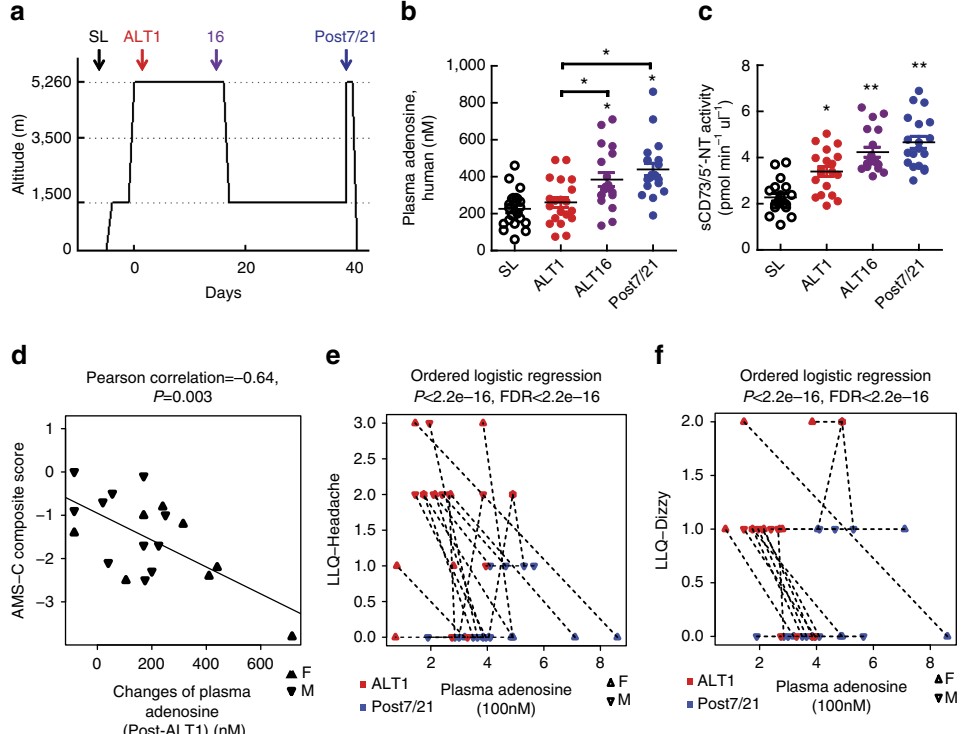

**Figure 1 | Elevated circulating purinergic components are associated with acclimatization and subsequent hypoxic adenosine response upon re-ascent.**
(**a**) Schematic illustration of Human AltitudeOmics Study. (**b**) Plasma adenosine levels of samples collected at different time points ($n = 21$, error bar: mean ± s.e.m., *$P < 0.05$, $t$-test). (**c**) sCD73 activities of plasma samples collected at different time points ($n = 21$, error bar: mean ± s.e.m., *$P < 0.05$, **$P < 0.001$, $t$-test). (**d**) Changes of plasma adenosine was significantly correlated with improved acclimatization judged by negative correlation with changes of AMS-C composite scores (acute mountain sickness; $n = 21$, Pearson correlation $r = -0.64$, $P = 0.003$). (**e,f**) Plasma adenosine is significantly associated with improved acclimatization at Post7/21 in comparison with ALT1 (**e**) associated with LLQ-Headache (ordered logistic regression, $n = 21$, $P < 2.2e - 16$, FDR $< 2.2e - 16$), (**f**) associated with LLQ-Dizzy (ordered logistic regression, $n = 21$, $P < 2.2e - 16$, FDR $< 2.2e - 16$). FDR = false discovery rate.

observe a significant correlation between increased sCD73 activity with decreased AMS-C composite score on Post7/21 compared with ALT1 (Supplementary Fig. 2). These data support our hypothesis that the rapid rise in plasma adenosine upon re-ascent represents a physiological retention for rapid acclimatization upon re-ascent.

Because elevated plasma adenosine was associated with reduced AMS-C composite score upon re-ascent, we used ordered logistic regression to further assess which specific physiological outcomes measured in the AMS-C composite score were most significantly associated with plasma adenosine levels on ALT1 and Post7/21, respectively. Among all of the categorical outcomes, including Lake Louise Questionnaire (LLQ)-Fatigue, LLQ-Headache, LLQ-Gastrointestinal, LLQ-Dizzy and LLQ-AMS Score in the AMS-C composite score, we found that LLQ-Headache (ordered logistic regression, $P < 2.2e - 16$, false discovery rate (FDR) $< 2.2e - 16$) and LLQ-Dizzy (ordered logistic regression, $P < 2.2e - 16$, FDR $< 2.2e - 16$) were two key physiological parameters measured in AMS-C composite score that significantly associated with plasma adenosine levels on ALT1 and Post7/21, respectively (Fig. 1e,f). Overall, our human studies revealed that increased plasma adenosine levels are correlated to initial acclimatization, and that higher levels of plasma adenosine are correlated to the rapid acclimatization upon re-ascent.

**eENT1 plays a major role in extracellular adenosine uptake.** Circulating adenosine levels are governed by synthesis via CD73, degradation by ADA and cellular uptake by facilitated ENTs. However, the function and regulation of ENTs under acute

hypoxia remains undetermined. To examine the role of ENTs in regulating extracellular adenosine, we translated our human finding to *in vivo* studies with intact animals to determine the role of erythrocytes in the uptake of plasma adenosine. Specifically, we used an *in vivo* $C^{14}$-adenosine uptake assay to accurately trace the distribution of adenosine in mice at different time points. We found that injected $C^{14}$-adenosine was rapidly transported inside erythrocytes, as quickly as 1 minute (min) (Fig. 2a). In addition, we found that the distribution of $C^{14}$-adenosine among whole blood, plasma and red blood cells (RBCs) reached a maximum within 1 min and maintained steady levels for at least 10 min (Fig. 2a). We chose to compare the distribution of $C^{14}$-adenosine at 5 min after injection. We found that major portion of injected $C^{14}$-adenosine was retained in the circulation or whole blood (68%). Hence, ~30% of total injected adenosine was taken up by peripheral tissues. Intriguingly, ~54% of injected adenosine was taken up by erythrocytes, which is responsible for almost 90% of injected adenosine in whole blood cells. Approximately 10% of injected adenosine was taken up by other non-erythrocyte circulating cells and 5% of total injected adenosine was retained in plasma (Fig. 2a). Thus, these results indicate that erythrocytes are the major cell type responsible for the rapid uptake of extracellular adenosine *in vivo*.

Next, to investigate which ENT on erythrocytes is the predominant adenosine transporter, we carried out *in vitro* adenosine uptake assays using erythrocytes isolated from wild-type (WT C57BL/6) mice and normal human subjects in the presence or absence of a specific ENT1 inhibitor, nitrobenzylmercaptopurine (NBMPR), or an ENT general inhibitor,

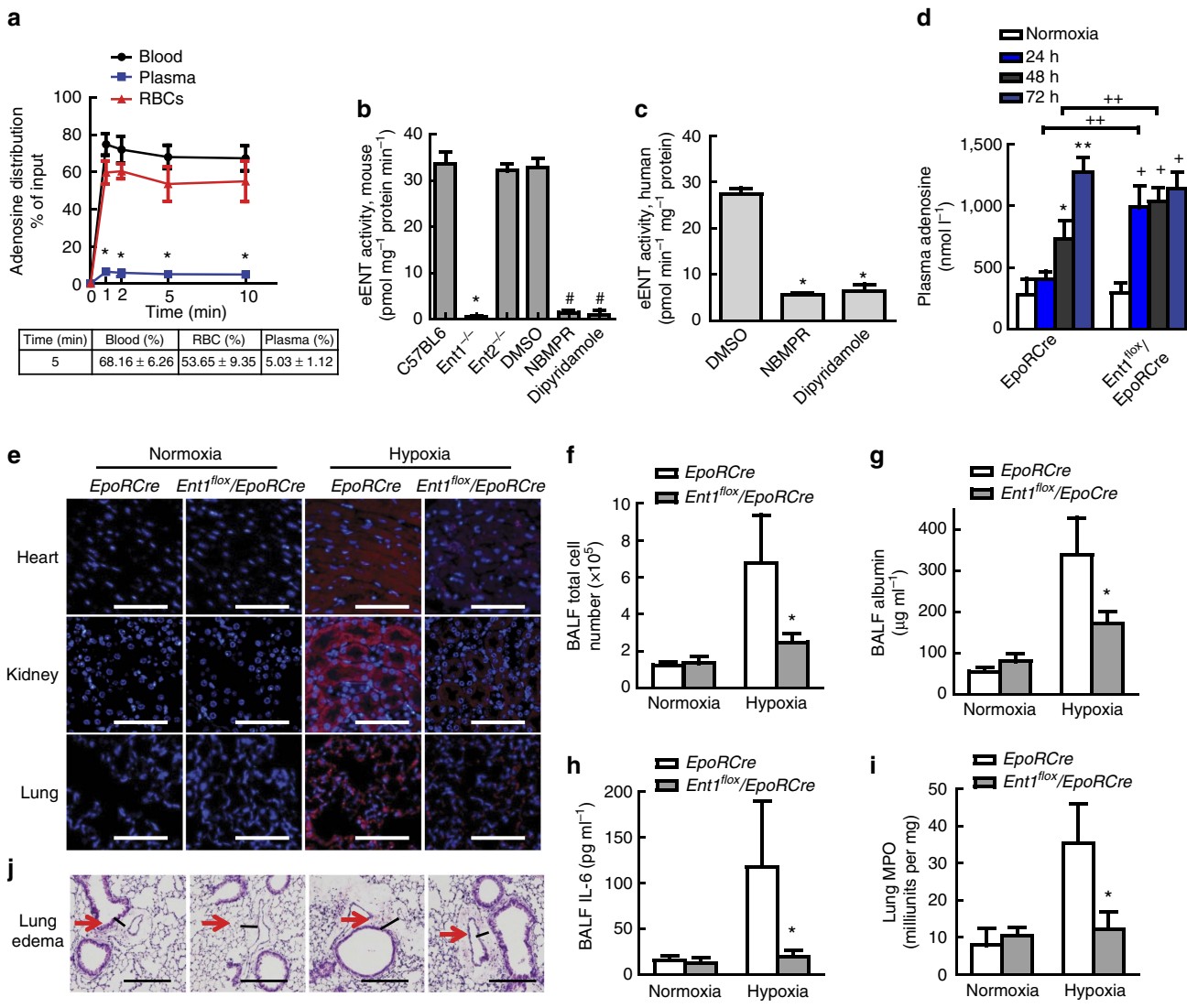

**Figure 2 | eENT1 is the major factor eliminating extracellular adenosine.** (**a**) *In vivo* adenosine uptake assay, comparison is between plasma and blood, $n = 4$, $P < 0.05$. (**b**) *In vitro* adenosine uptake assay, *Comparison between RBC from $ENT1^{-/-}$ and WT, $n \geq 4$, $P < 0.001$; #comparison between dipyridamole or NBMPR and DMSO, $n = 4$, $P < 0.001$. (**c**) *In vitro* adenosine uptake assay using human erythrocytes, $n = 4$, *$P < 0.05$. (**d**) Plasma adenosine from hypoxia-treated $Ent1^{flox/flox}/EpoRCre$ or $EpoRCre$ mice. *: between normoxia and 48-h hypoxia, $n = 4$, $P < 0.05$. **between 24- and 72-h hypoxia, $n = 4$, $P < 0.05$. $^{+}$ between normoxia and 24-, 48- or 72-h hypoxia, $n = 4$, $P < 0.05$. $^{++}$ between $EpoRCre$ and $Ent1^{flox/flox}/EpoRCre$ after 48- or 72-h hypoxia, $n = 4$, $P < 0.05$. (**e**) Reduced tissue hypoxia was observed in $Ent1^{flox/flox}/EpoRCre$ compared with $EpoRCre$ after 72-h hypoxia treatment by HypoxiaProbe, red: HypoxiaProbe, blue: DAPI; scale bar, 200 µm. (**f**–**h**) Deletion of eENT1 protects from acute lung injury indicated by (**f**) total cell number ($n = 4$, *$P < 0.05$), (**g**) albumin ($n = 4$, *$P < 0.05$) and (**h**) IL-6, ($n = 4$, *$P < 0.05$) in BALF from 72-h hypoxia-treated animals. (**i**) Pulmonary neutrophil accumulation by MPO assay in lung tissues from 72-h hypoxia-treated animals ($n \geq 4$, *$P < 0.05$). (**j**) H&E-stained lung section from 72-h hypoxia-treated animals (edema, red arrows); scale bar, 500 µm. All error bars: mean ± s.d. and *t*-test.

dipyridamole. NBMPR inhibits >95% of ENT1 activity at 0.1 µM and has no effects on other ENTs at this concentration[26,27]. NBMPR and dipyridamole treatment significantly inhibited eENT activity in mouse and human compared with untreated cells (Fig. 2b,c). Moreover, there was no significant difference between the NBMPR- and dipyridamole-treated groups (Fig. 2b,c). Consistent with these pharmacological studies, adenosine uptake in the erythrocytes isolated from global ENT1-deficient mice ($Ent1^{-/-}$) was almost completely abolished (Fig. 2b). In contrast, erythrocytes isolated from global $Ent2^{-/-}$ mice maintained similar ENT transporter activity as normal mouse erythrocytes (Fig. 2b). These results indicate that ENT1 on the erythrocytes is the major ENT responsible for uptake of extracellular adenosine in mice and humans.

**eENT1 regulates extracellular adenosine under acute hypoxia.** To evaluate the importance of erythrocyte equilibrative nucleoside transporter 1 (eENT1) in the uptake of adenosine *in vivo*, we generated erythrocyte-specific ENT1 and ENT2 knockouts by crossing $Ent1^{flox/flox}$ or $Ent2^{flox/flox}$ with $EpoRCre$-GFP ($EpoRCre$) mice (Supplementary Fig. 3a)[28]. Western blot analysis showed that most of the ENT1 or ENT2 protein was absent from the erythrocytes of $Ent1^{flox/flox}/EpoRCre$-GFP ($Ent1^{flox/flox}/EpoRCre$) or $Ent2^{flox/flox}/EpoRCre$-GFP ($Ent2^{flox/flox}/EpoRCre$) mice, respectively, but not in other tissues, such as the brain (Supplementary Fig. 3b). Moreover, we demonstrated that ENT transport activity was significantly decreased in the erythrocytes isolated from $Ent1^{flox/flox}/EpoRCre$ mice compared with $EpoRCre$ mice, but not the erythrocytes from $Ent2^{flox/flox}/EpoRCre$ mice

(Supplementary Fig. 3c). These results indicate that we successfully deleted eENT1 or eENT2 in mice.

To determine the importance of eENT1 in the acute hypoxic adenosine response, we exposed *EpoRCre* (controls) and *Ent1^flox/flox/EpoRCre* mice to hypoxia (8% oxygen, equivalent to an altitude of 7,500 m) for up to 72 h. Similar to human high-altitude studies, plasma adenosine levels were induced by hypoxia in both *EpoRCre* control and *Ent1^flox/flox/EpoRCre,* while it increased much faster in *Ent1^flox/flox/EpoRCre* than *EpoRCre* mice (Fig. 2d). Of note, we found that plasma adenosine levels were induced to higher levels in mice than in humans under hypoxia (Figs 1b and 2d). This difference is likely due to the mice being exposed to 8% normobaric hypoxia, equivalent to an altitude of 7,500 m, which is higher than humans who were exposed to 5,260 m equivalent to 10% hypobaric hypoxia. These studies indicate that eENT1 plays an important role in the rapid elevation of extracellular adenosine under acute hypoxia in mice.

Next, we assessed tissue hypoxia using HypoxiaProbe (pimonidazole, forming protein adducts in hypoxic cells when $pO_2 < 10$ mm Hg) in mice[29]. A significant decrease in tissue hypoxia in the heart, kidney and lung was observed in hypoxia-exposed *Ent1^flox/flox/EpoRCre* mice compared with *EpoRCre* control mice, indicating that rapid induction of extracellular adenosine is likely beneficial to prevent tissue hypoxia (Fig. 2e images; statistical results in Supplementary Fig. 4a–c). Because lung damage is known to be induced by acute hypoxia[30], we chose to further quantify pulmonary vascular leakage and inflammation in those mice. We found significantly reduced total cell numbers, albumin and interleukin (IL)-6 concentration in collected bronchoalveolar lavage fluid (BALF) in *Ent1^flox/flox/EpoRCre* than *EpoRCre* mice under hypoxia (Fig. 2f–h). Moreover, pulmonary myeloperoxidase (MPO) activity, a specific enzyme for primary neutrophils[31] was significantly less induced in hypoxia-challenged *Ent1^flox/flox/EpoRCre* compared with *EpoRCre* mice (Fig. 2i). Finally, haematoxylin and eosin (H&E) staining showed a significant increase in perivascular fluid accumulation (oedema) in hypoxia-treated *EpoRCre* mice compared with *Ent1^flox/flox/EpoRCre* (Fig. 2j, red arrows; Supplementary Fig. 4d). Our findings indicate that the loss of eENT1 leads to quicker and higher accumulation of extracellular adenosine under acute hypoxia resulting in reduced hypoxia-induced tissue inflammation and damage.

**Hypoxia induces eENT1 ubiquitination and degradation.** To our surprise, we observed that extracellular adenosine levels in control mice eventually accumulated to levels similar to that of the eENT1-deficient mice after 72-h hypoxia (Fig. 2d). These findings raise an intriguing possibility that 72-h hypoxia exposure likely reduces eENT1 activity and thus allows the accumulation of extracellular adenosine in the control mice to a similar level as in eENT1-deficient mice. To test this possibility, we measured eENT activity in control mice. We found that under hypoxia, eENT activity was significantly reduced at 48 h and further reduced at 72 h (Fig. 3a). We also compared sCD73 activity and plasma ATP levels between controls and erythrocyte-specific ENT1 knockouts under normoxia and 72-h hypoxia. We found that sCD73 activity and plasma ATP levels were induced by hypoxia compared with normoxia in both *EpoRCre* and *Ent1^flox/flox/EpoRCre* mice, with no significant difference in hypoxia-induced sCD73 activity and plasma ATP levels between them (Fig. 3b,c). Thus, in response to hypoxia, there is a coordinated increase in circulating sCD73 activity, plasma ATP levels and reduced ENT activity, which work together to promote a rapid rise in plasma adenosine.

Because erythrocytes have no nuclei, we speculated that reduced ENT1 transporter activity in response to hypoxia is likely due to reduced protein levels. We measured membrane ENT1 protein levels in the erythrocytes from *EpoRCre* mice at normoxia and under hypoxia for up to 72 h. Intriguingly, as with ENT activity, membrane-anchored ENT1 protein levels in the erythrocytes were reduced at 48 h and further reduced after 72-h hypoxia challenge (Fig. 3d, left panel; Supplementary Fig. 5). To further determine whether hypoxia-mediated reduced membrane eENT1 is through ubiquitin–proteasomal degradation, we performed immunoprecipitation (IP) followed with western blot analysis. Specifically, we pulled down ubiquitinated protein from erythrocyte lysates from normoxia- or hypoxia-treated animals using an antibody specific for ubiquitin, followed by western blot analysis against ENT1. We found that polyubiquitinated ENT1 $((Ub)_n$-ENT1)) levels in lysates were substantially elevated after 48-h hypoxia compared to normoxia and were degraded after 72-h hypoxia exposure (Fig. 3d, right panel; Supplementary Fig. 6). These results indicate that hypoxia-mediated reduced membrane eENT1 is due to ubiquitination-dependent proteasome degradation *in vivo*.

**Adenosine induces eENT1 degradation under acute hypoxia.** Given the fact that adenosine is a potent hypoxic-signalling sensor, we speculated that elevated sCD73-induced production of extracellular adenosine may function as a positively feed-forward loop to promote ubiquitination of eENT1, subsequent accumulation of extracellular adenosine to protect hypoxic tissue damages. To test this possibility, we took a genetic approach to expose WT and $Cd73^{-/-}$ mice to normoxia and hypoxia up to 72 h. Consistently, we found that hypoxia-induced plasma adenosine levels in WT mice but not $Cd73^{-/-}$ mice[19]. Moreover, we found that eENT activity and membrane-anchored eENT1 levels were significantly reduced by hypoxia in WT mice but not in $Cd73^{-/-}$ mice (Fig. 3e). These findings provide *in vivo* mouse genetic evidence that sCD73-dependent elevation of plasma adenosine underlies hypoxia-mediated downregulation of membrane-anchored eENT1 levels and thus reduced eENT1 activity.

Extracellular adenosine elicits multiple cellular functions by the activation of four adenosine receptors[12,13]. To determine whether adenosine signalling via its specific receptors underlies reduction of eENT1, we compared eENT1 activity in primary cultured erythrocytes (mature and enucleated erythrocytes) isolated from WT and four adenosine receptor-deficient mice in response to 5′-*N*-ethylcarboxamidoadenosine (NECA) treatment, a non-metabolized adenosine analogue[32]. We found that NECA directly and significantly reduced eENT1 activity in WT mouse erythrocytes (Fig. 3f). However, genetic deletion of *Adora2b* but not *Adora1*, *Adora2a* or *Adora3* attenuated NECA-mediated reduction of ENT activity (Fig. 3f). Similar to $CD73^{-/-}$ mice, we confirmed our *in vitro* studies *in vivo* by showing that hypoxia-mediated reduction of membrane-anchored eENT1 and ENT activity were abolished in ADORA2B-deficient mice (Fig. 3e). Here we provided *in vivo* and *in vitro* genetic evidence that CD73-dependent production of plasma adenosine signalling via ADORA2B underlies hypoxia-mediated reduction of ENT activity by decreased membrane-anchored ENT1 levels in erythrocytes.

ADORA2B is a Gs-coupled receptor and PKA activation is reportedly involved in ubiquitination[33]. Thus, it is possible that NECA-mediated reduction of eENT1 transporter activity and protein level is dependent on PKA-mediated phosphorylation and subsequent ubiquitination. To test this possibility, we first pulled down phosphorylated PKA substrate from erythrocyte lysates from normoxia- or hypoxia-treated animals using an antibody specific for phospho-PKA substrate, followed by western blot against ENT1 (Fig. 4a; Supplementary Fig. 7). We found that

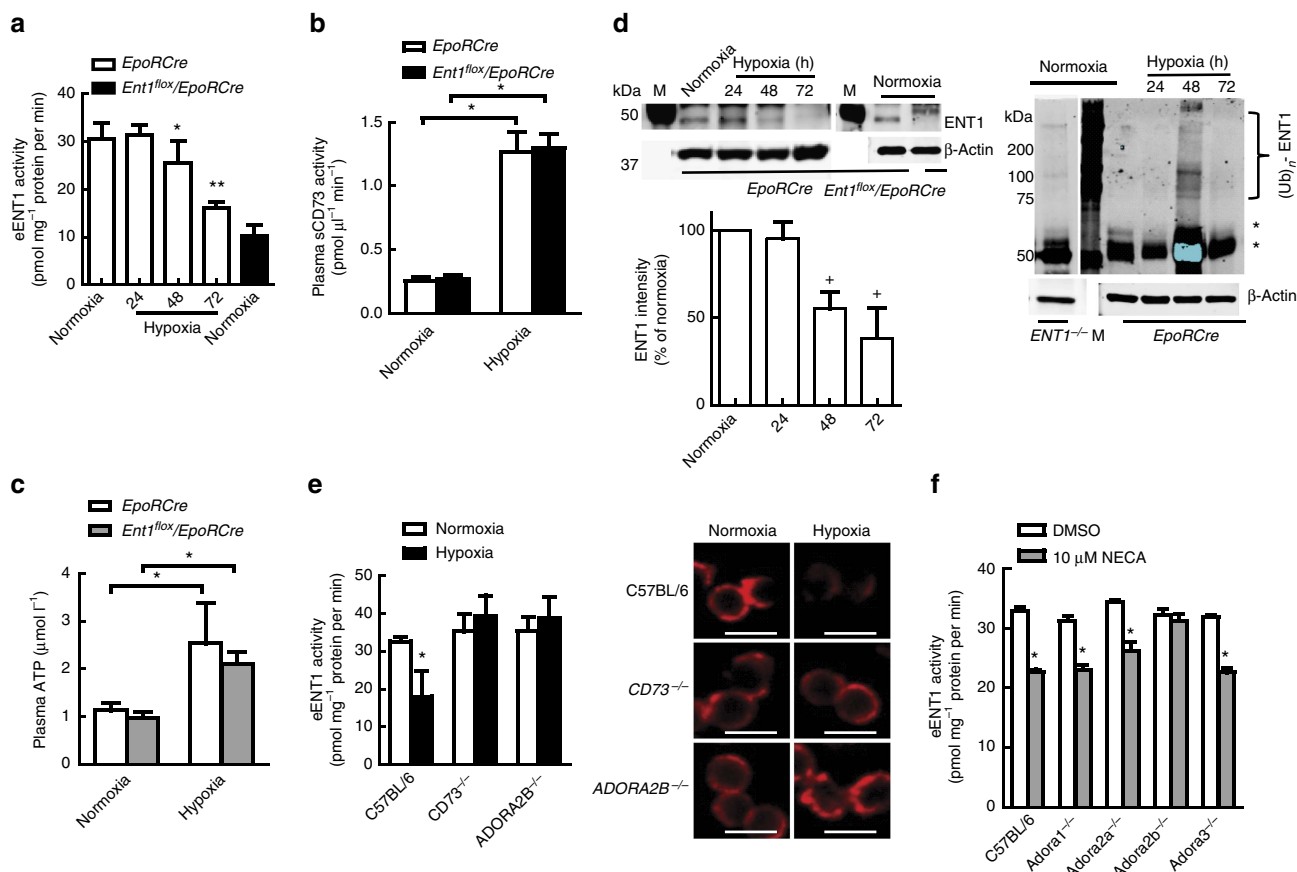

**Figure 3 | Hypoxia downregulates eENT1 through adenosine–ADORA2B–ubiquitin signalling.** (**a**) Hypoxia decreased eENT1 activity in erythrocytes from hypoxia-treated animals as judged by *in vitro* adenosine uptake assay, $n \geq 4$, *comparison between normoxia and 48-h hypoxia-treated animals, $P < 0.05$; **comparison between normoxia and 72-h hypoxia-treated animals, $P < 0.001$, error bar: mean ± s.e.m. (**b**) sCD73 activity in *Ent1^flox/flox^/EpoRCre* or *EpoRCre* mice with or without hypoxia treatment, $n = 4$, error bar: mean ± s.d., $P < 0.001$, *t*-test. (**c**) Plasma ATP in *Ent1^flox/flox^/EpoRCre* or *EpoRCre* mice with or without hypoxia treatment, $n = 4$, error bar: mean ± s.e.m., $P < 0.05$. (**d**) Erythrocyte ENT1 is reduced during hypoxia treatment through ubiquitination by western blot against ENT1 (left panel, β-actin was used as loading control, $n = 3$, error bar: mean ± s.d., $^+P < 0.05$). Right panel, IP ubiquitin followed by western blot against ENT1, *non-specific bands. (**e**) Hypoxia-mediated downregulation of eENT1 activity (left panel) and eENT1 protein (immunofluorescence against ENT1, red colour: ENT1, right panel; scale bar, 5 μm) were observed in C57BL/6 mice, but not *CD73^−/−^* or *AODRA2B^−/−^* mice when we treated these animals under hypoxia (8% oxygen) for 72 h, $n \geq 6$, error bar: mean ± s.d., $P < 0.05$. (**f**) Adenosine signalling through ADORA2B downregulates eENT1 activity as judged by *in vitro* adenosine uptake assay ($n = 3$, error bar: mean ± s.d., *$P < 0.05$). *t*-test is used for all comparisons in this figure.

hypoxia treatment increased ENT1 phosphorylation by PKA and reached a peak at 48 h, then decreased after 72-h hypoxia treatment that was the same trend as ubiquitinated ENT1 (Fig. 3d, right panel). Next, we treated primary cultured WT mouse erythrocytes with or without Bay60-6583, a specific ADORA2B agonist[34], or forskolin, a specific PKA activator[35], in the presence or absence of two structurally independent inhibitors of the 20S proteasome, the proteasome found in erythrocytes[36], MG132 (ref. 37) or Bortezomib[38], or a PKA inhibitor, H-89 (ref. 39). We found that H-89, MG132 and Bortezomib significantly attenuated Bay60-6583-mediated decrease in eENT1 activity in a dosage-dependent manner, respectively (Fig. 4b). Moreover, forskolin alone directly downregulated eENT1 activity and this downregulation was also attenuated by H-89, MG132 and Bortezomib in a dose-dependent manner (Fig. 4c). Finally, we also monitored ENT1 localization and ubiquitinated proteins following Bay60-6583 or forskolin treatment using immunofluorescence (IF) confocal image analysis. Both Bay60-6583 and forskolin treatments induced ubiquitination (green fluorescence) and subsequently ENT1 translocation from membrane to cytosol

(red fluorescence), and finally degradation in a time-dependent manner. These events were blocked by Bortezomib or H-89 treatment (Fig. 4d). Notably, Bortezomib blocked eENT1 degradation but not translocation and H-89 blocked both ENT1 translocation and degradation, indicating that PKA-mediated phosphorylation is required for ADOAR2B-induced ubiquitination and subsequent proteasome degradation of ENT1 (Fig. 4d). Taken together, we provide both genetic and pharmacological *in vivo* and *in vitro* evidence that PKA functions downstream of ADOAR2B underlying adenosine-mediated reduction of eENT1 activity by inducing its phosphorylation, ubiquitination and proteasome degradation.

**PKA mediates eENT1 degradation in response to high altitude.** To validate our mouse studies in humans, we measured PKA phosphorylated ENT1, ubiquitinated ENT1 and ENT1 protein levels in erythrocytes isolated from the human volunteers participating in the AltitudeOmics study. We found that PKA-mediated phosphorylated ENT1 started to rise on ALT1, reached the peak on ALT7, began to decrease and maintained at

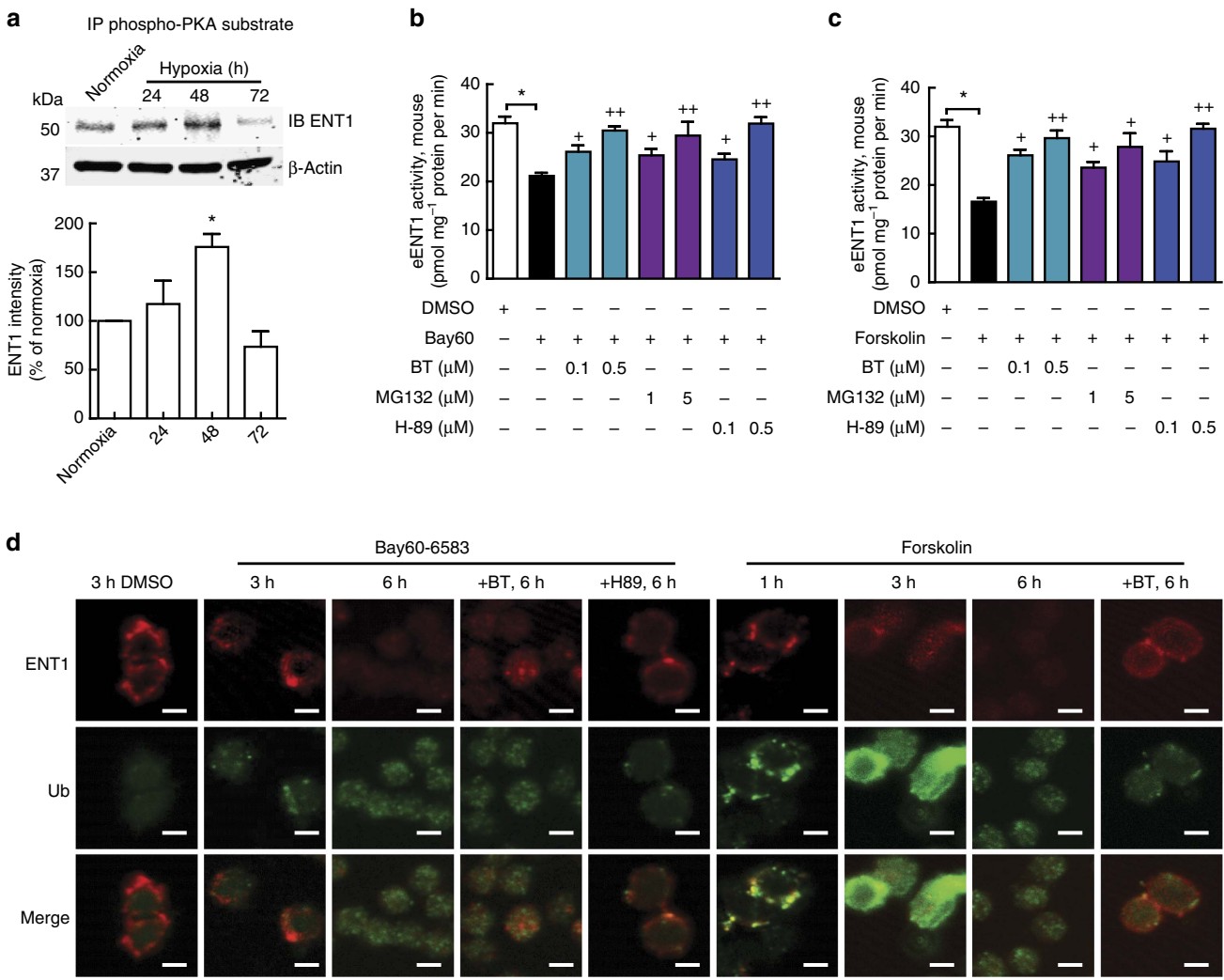

**Figure 4 | ADORA2B–PKA–ubiquitin–proteasomal degradation of eENT1 in mice.** (**a**) IP–western blot using phospho-PKA subtract antibody pull-down proteins from RBC lysate followed by western blot against ENT1, β-actin as an input control, $n = 3$, *compared with normoxia, $P < 0.05$, $t$-test. (**b**,**c**) *In vitro* adenosine uptake assay. Activation of ADORA2B by Bay 60-6583 (B60) or activation of PKA by forskolin decreases eENT1 activity that can be blocked by H-89, MG132 or BT (Bortezomib). (**b**) By B60, $n = 3$, *comparison between DMSO and B60 treatment, $P < 0.05$; $+$ and $++$ comparison between B60 and B60 plus inhibitor, $+P < 0.05$, $++P < 0.001$. (**c**) By forskolin, $n = 3$, *comparison between DMSO and forskolin treatment, $P < 0.05$; $+$ and $++$ comparison between forskolin and forskolin plus inhibitor, $+P < 0.05$, $++P < 0.001$. (**d**) Immunofluorescence, activation of ADORA2B or PKA leads to eENT1 ubiquitination and degradation. B60 and forskolin induced protein ubiquitination in a time-dependent manner and the effect of B60 can be blocked by H-89 (middle panels). B60 and forskolin induced eENT1 translocation to the cytosol after 3-h treatment and degradation after 6-h treatment. H-89 blocked B60-induced translocation and degradation of eENT1. BT blocked B60- and forskolin-induced degradation of eENT1, but not the translocation (upper panels). Red: eENT1, green: ubiquitin, scale bar: 2 μm. Error bars: mean ± s.d., $t$-test.

similar but still elevated levels on ALT16 and on Post7 upon re-ascent compared with SL (Fig. 5a; Supplementary Fig. 8). Similar to the trend of PKA-mediated ENT1 phosphorylation, the polyubiquitination of ENT1 ((Ub)$_n$-ENT1)) in the cell lysate started to rise on ALT1, reached the peak on ALT7, began to decrease and maintained at similar but still elevated levels on ALT16 and on Post7 upon re-ascent compared with SL (Fig. 5b; Supplementary Fig. 9). Supporting the mouse finding of PKA-mediated phosphorylation-dependent ubiquitination and proteasomal degradation of ENT1, we found that membrane ENT1 protein levels were significantly reduced at ALT7, further reduced on ALT16 and remained a similar low level upon re-ascent at Post7 (Fig. 5c; Supplementary Fig. 10). These findings provide strong human *in vivo* evidence that high-altitude hypoxia leads to PKA-mediated phosphorylation and ubiquitination-dependent proteasome degradation of eENT1, and that reduced eENT1 membrane protein is retained upon re-ascent. As such, we

observed a much more rapid elevation of extracellular adenosine upon re-ascent and concurrent retention of physiological acclimatization (Fig. 1).

**Hypoxic adenosine response is through eENT1 downregulation.** Here we revealed that ENT1 protein in human erythrocytes is reduced by initial hypoxia-induced proteasome degradation and remained low upon re-ascent to high altitude. Because erythrocytes contain no nuclei and do not carry out translation, they lack the ability to make new protein. Thus, our human *in vivo* high-altitude studies and mouse genetic studies support a novel but compelling hypothesis that erythrocytes, as the most abundant cells in our body, retain a hypoxic adenosine response for a certain time period until hypoxia-exposed RBCs are replaced by newly synthesized RBCs containing higher levels of ENT1 protein. To test this intriguing possiblity, we mimicked human high-altitude studies by

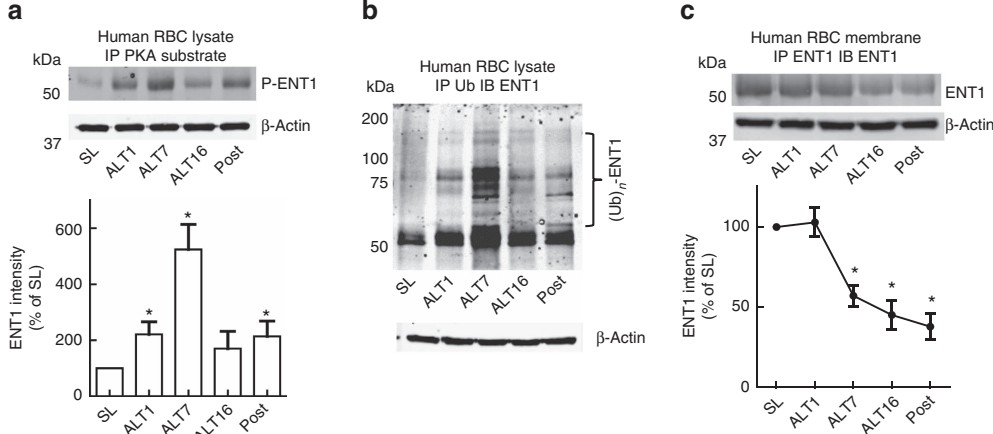

**Figure 5 | PKA phosphorylation-mediated ubiquitination and porteasomal degradation of eENT1 in humans at high altitude and upon re-ascent. (a)** IP phospho-PKA substrate followed by western blot against ENT1, (IP–western blot is representative image from three experiments using three sets of RBC samples from the Human AltitudeOmics Study), P-ENT1: PKA phosphorylated ENT1. **(b)** IP ubiquitin followed by western blot against ENT1, $n = 3$, (IP–western blot was performed using RBC samples from three subjects), *: compared with SL, error bars: mean ± s.d., $P < 0.05$, t-test. **(c)** Hypoxia-induced downregulation of eENT1 protein levels in erythrocyte samples judged by western blot against ENT1, $n = 3$ (RBC samples from three subjects), *compared with SL, error bars: mean ± s.d., $P < 0.05$, t-test.

exposing WT mice to hypoxia, bringing them to normoxia and re-exposing them to hypoxia (Fig. 6a). First, to trace ENT1 protein levels and activity in erythrocytes from first hypoxia to second hypoxia, we *in vivo* labelled erythrocytes at 24 h before first hypoxia by *N*-hydroxysuccinimide (NHS) biotin[20]. Because we found that 72-h hypoxia is long enough to induce eENT1 proteasomal degradation and reduce its activity in WT mice but not to significantly increase in erythropoiesis, we exposed WT mice to hypoxia for 72 h (8% oxygen balanced with nitrogen; Fig. 3a,d). After the first hypoxic challenge, mice were returned to normoxia (21% oxygen) for 3 days up to 50 days because the life span of mouse erythrocytes is 55 days. At the end of each time point, some of the mice were re-exposed to hypoxia for 24 h (at Post3 or Post50) and then killed for further analysis (Fig. 6a, illustration).

As expected, flow cytometry analysis of biotinated erythrocytes showed that they started to decrease on Post3 and almost completely disappeared by Post50, indicating that old labelled erythrocytes were replaced by newly synthesized erythrocytes by Post50 (Fig. 6b). Follow-up co-immunoflurosence staining of eENT1 and biotin showed that membrane-anchored eENT1 staining was significantly reduced after first hypoxic challenge (72-h hypoxic challenge; Fig. 6c, upper panel). Moreover, eENT1 membrane protein was still reduced to the similar level at 72 h following first hypoxia in the biotinated erythrocytes after mice were brought back to normoxia (Post3 normoxia group) and followed by 1 day exposure to second hypoxia (Post3 re-hypoxia group; Fig. 6c). In contrast, eENT1 protein levels almost returned to the normal levels in the non-biotinylated and newly synthesized erythrocytes in mice after 50 days of normoxia (Post50 normoxia) and followed by 1 day exposure to second hypoxia, indicating that old erythrocytes had been replaced by newly synthesized erythrocytes and thus eENT1 expression levels were back to normal levels (Fig. 6c, Post50 re-hypoxia). Functionally, we validated that the eENT activities also followed the same trend as eENT1 protein levels (Fig. 6d). Similar to human high-altitude studies, plasma adenosine levels were rapidly induced to significantly higher levels after 1 day second hypoxia following 3 days at normoxia (Fig. 6e, Post3). In contrast, the faster and higher increases of plasma adenosine during second hypoxia challenge went away when the mice were brought back to normoxia for 50 days and re-exposed to hypoxia (Fig. 6e, Post50).

Taken together, our mouse findings recapitulate human high altitude observations and provide strong evidence that erythrocytes retain a hypoxic adenosine response through downregulation of eENT1 protein and activity until hypoxia-challenged erythrocytes are replaced by newly synthesized erythrocytes.

## Discussion

Here we report both human *in vivo* high-altitude studies and mouse genetic hypoxic findings that eENT1 is a key purinergic-signalling component responsible for hypoxia-induced plasma adenosine. Mechanistically, we revealed that increased plasma adenosine subsequently signals via ADORA2B, inducing PKA-dependent phosphorylation, ubiquitination and proteasomal degradation of eENT1 protein mediated by the initial hypoxia challenge and in turn results in decreased uptake of extracellular adenosine and accumulation of plasma adenosine. The reduced eENT1 resulting from the initial hypoxic exposure establishes an erythrocyte hypoxic adenosine response for a second hypoxic exposure, and underlies a faster and improved acclimatization upon re-ascent associated with high levels of circulating adenosine (Fig. 7).

Ascending to high altitude is challenging due to limited oxygen supply. Consequently, an acute and integrative physiological response to protect against hypoxic tissue damage is essential to allow adaptation and survival. Faster and improved acclimatization is a consistent and compelling observation seen in humans upon re-ascent. Although elevation of adenosine is widely considered beneficial under acute hypoxic settings, changes in plasma adenosine during high-altitude acclimatization and upon re-ascent were unrecognized until we measured circulating purinergic components in healthy human lowland volunteers involved in the AltitudeOmics study[5]. Here we demonstrated that sCD73 activity and plasma adenosine levels were concurrently induced by initial exposure to high altitude and rapidly accumulated to much higher levels upon re-ascent following 7 days at a much lower elevation. Functionally, plasma adenosine levels were positively associated with initial altitude acclimatization and subsequent rapid acclimatization upon re-ascent.

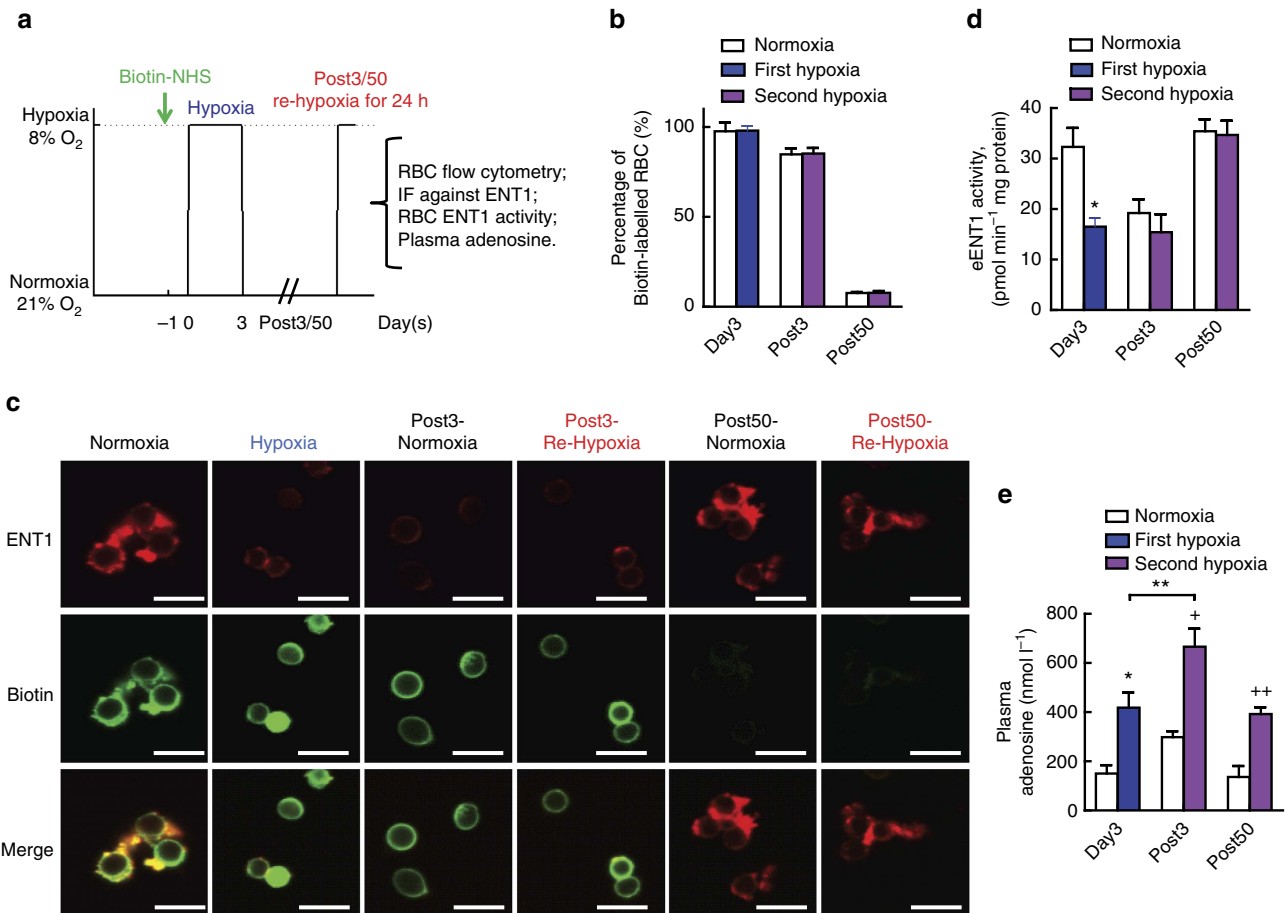

**Figure 6 | Erythrocyte has hypoxic adenosine response through hypoxia-mediated downregulation of eENT1.** (**a**) Experimental illustration for acclimatization hypoxic adenosine response. (**b**) Erythrocyte life span was indicated by biotin NHS labelling and revealed by flow cytometry. (**c**) eENT1 changes during hypoxia treatment as judged by immunofluorescence, red: ENT1, green: Biotin; scale bar, 5 μm. (**d**) eENT1 activity ($n = 4$, *$P < 0.05$). (**e**) Plasma adenosine levels. $n = 4$ for all groups, *between normoxia and first hypoxia, $P < 0.05$; +between Post3 normoxia and Post3 re-hypoxia, $P < 0.05$; ++between Post50 normoxia and Post50 re-hypoxia, $P < 0.05$; **between first hypoxia and Post3 re-hypoxia, $P < 0.05$. Error bars: mean ± s.d., $t$-test.

The erythrocyte is the most abundant circulating cell type and it has been speculated that erythrocytes regulate extracellular adenosine uptake[40]. The ENT (SLC29) family contains four members, namely, ENT1 through ENT4 that are widely expressed in mammalian tissue with different expression levels[41]. Early studies demonstrated that the inhibition of ENT1 is beneficial for acute liver and kidney damage induced by ischaemia and reperfusion in experimental models[42]. However, the function of eENT1 in response to hypoxia could not be determined without our newly developed genetically manipulated mice with erythrocyte-specific ablation of ENT1 or ENT2. These mice represent important genetic tools used to demonstrate that ENT1 but not ENT2 on the erythrocyte plays a key role in adenosine uptake. The ablation of eENT1 but not ENT2 in mice results in much faster and higher accumulation of plasma adenosine under acute hypoxia. We further demonstrated that genetic deletion of eENT1 results in the rapid accumulation of extracellular adenosine and is beneficial to protect against acute hypoxia-induced tissue damage including vascular leakage and inflammation in the lung tissues compared with the control mice. Taken together, we have provided both human and mouse evidence that eENT1 is critical to uptake exogenous extracellular adenosine and that reduced eENT1 is a previously unrecognized regulatory mechanism to enhance adenosine hypoxia response to counteract tissue hypoxia.

Multiple molecules including hypoxia-inducible factor-1α and nitric oxide are associated with physiological and pathological hypoxic responses[43,44]. Of note, hypoxia-inducible factor-1α regulates transcription of multiple purinergic-signalling components including CD73, ADOAR2B, ADORA2A and ENTs[45–47]. Recent studies showed that alveolar ADORA2B crosstalks with ENT2 to dampen acute lung injury[48]. However, erythrocytes have no nuclei and therefore regulation of eENT1 is likely independent of transcription and translation. Here we demonstrated that erythrocyte ENT1 is regulated by ubiquitination-dependent proteasomal degradation machinery in humans under high altitude and mice under hypoxia. Thus, the hypoxia-mediated degradation of eENT1 lowered plasma adenosine uptake and promoted the accumulation of extracellular adenosine. To our surprise, we found that elevated sCD73-induced production of plasma adenosine signalling via ADOAR2B-induced ubiquitination and degradation of ENT1 in both human and mouse erythrocytes. Moreover, we revealed that PKA-mediated phosphorylation of ENT1 is required for ADOAR2B-mediated ubiquitination and degradation of ENT1 in both human and mouse erythrocytes. Among four adenosine receptors, ADORA2B has a lowest affinity and it is usually activated by adenosine at micromolar concentration[49]. Thus, erythrocyte ADORA2B-mediated PKA phosphorylation and ubiquitination of ENT1 do not occur until adenosine is elevated

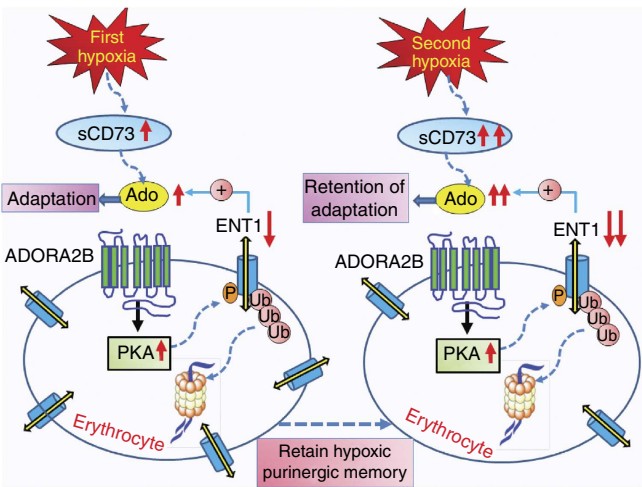

**Figure 7 | Working model.** Our findings support a working model that elevated sCD73-mediated increased adenosine signalling via ADORA2B results in PKA-mediated phosphorylation, ubiquitination and proteasomal degradation of ENT1 in the erythrocytes of humans at high altitude and mice exposed to hypoxia. Thus, purinergic-signalling components play an important role in enhancing accumulation of plasma adenosine and thus promote initial adaptation to hypoxia in a feed-forward manner. Moreover, these newly identified erythrocyte purinergic components retain hypoxic adenosine response to promote quicker and higher elevation of plasma adenosine and thereby allow for rapid acclimatization upon re-ascent. Our findings offer multiple innovative therapies to counteract hypoxic challenge under physiological and pathological conditions.

to the micromolar range following 48 h of hypoxia in mice and day 7 at high altitude in humans. As such, CD73-ADORA2B-mediated erythrocyte proteasomal degradation of ENT1 functions synergistically as a feed-forward purinergic-signalling component to adapt to hypoxia by promoting adenosine production and reducing its elimination and thus enhancing accumulation of adenosine in the plasma. Moreover, besides erythrocytes, multiple-nucleated cells including endothelial cells also uptake extracellular adenosine[50]. Our finding about adenosine signalling via ADORA2B reducing eENT1 activity is likely not limited to erythrocytes and may also regulate ENT1 activity and extracellular adenosine in other cell types. This notion is strongly supported by a recent study showing that ADORA2A modulates drug efflux transporter, P-glycoprotein in the blood–brain barrier by ubiquitination[51]. Thus, our current findings add a new function for erythrocytes in the hypoxic adenosine response and reveal novel mechanisms regulate purinergic-signalling components.

Because erythrocytes do not have nuclei, hypoxia-induced proteasomal degradation of ENT1 in erythrocytes exposed to hypoxia cannot be replenished by new protein synthesis until new erythrocytes are generated. Moreover, we found that sCD73 activity was also further induced upon re-ascent to high altitude. Thus, the initial hypoxia-mediated elevation of sCD73 activity in the circulation and the proteosomal degradation of eENT1 work collaboratively to retain hypoxic adenosine response for acclimatization to high altitude resulting in a higher and quicker elevation of plasma adenosine and faster adaptation to second hypoxia. Extending from these findings, we further revealed that eENT1 contributes to initial adaption and retains a hypoxic adenosine response upon re-ascent until newly synthesized RBCs containing high ENT1 levels replace old RBCs with reduced ENT1-levels mediated by initial hypoxia. Thus, erythrocyte hypoxic purinergic memory is transient and is gradually lost when the newly synthesize erythrocytes replace

the old ones. These findings immediately suggest that targeting CD73, ENT1 and ADORA2B will likely enhance the adenosine hypoxic response and prolong this response in normal individuals ascending to high altitude for quicker acclimatization, in athletes with strenuous exercise to improve performance for gold medals and patients frequently facing hypoxia for better adaptation.

In conclusion, very little was known about the function and regulation of the erythrocyte hypoxic adenosine response before our current studies. Here we provide both human and mouse genetic evidence that CD73-depedent elevation of plasma adenosine signalling via ADORA2B-mediated phosphorylation, ubiquitination and degradation of ENT1 in the erythrocytes is a novel feed-forward signalling network underlying initial hypoxic adaptation and retention upon re-exposure to enhance the hypoxic adenosine response (Fig. 7). These findings add significant new insight to our understanding of molecular mechanisms of adaptation to hypoxia under physiological and pathological conditions and thereby open up novel therapeutic avenues to treat and prevent hypoxia.

## Methods

**Human study.** The current study was a part of the human AltitudeOmics project examining the integrative physiology of human responses to high-altitude hypoxia[5]. In brief, all procedures conformed to the Declaration of Helsinki and are approved by the Universities of Colorado and Oregon Institutional Review Boards and the US Department of Defense Human Research Protection office. All participants are healthy (mean ± s.d. age, 21 ± 1 years; stature, 1.78 ± 0.10 m; body mass, 69 ± 11 kg), non-smokers, cardiorespiratory disease free, born and raised at <1,500 m, and had not travelled to elevations >1,000 m in 3 months before the investigation[52]. Blood samples from each subject were drawn near SL at Eugene, OR, USA (baseline, 130 m, Barometric pressure = 749 mm Hg) and also on the first (ALT1), seventh (ALT7) and sixteenth (ALT16) day at Mt Chacaltaya, Bolivia (5,260 m, mean Barometric pressure = 406 mm Hg) and upon reasceding to high altitude for 1 day after descending to low altitude at 1,525 m for one (Post7, $n = 14$) or 3 weeks (Post21, $n = 7$; Fig. 1a). The detail experimental protocols were described previously[5].

AMS was accessed by Environmental Symptom Questionnaire (AMS-C) and LLQ[5]. AMS-C composite score is a self-reported 11-question inventory and indicative of AMS was considered when a score ≥0.7 (ref. 5). LLQ is a self-reported assessment of AMS symptoms.

**Mouse strains.** All animal protocols were in accordance with the University of Texas, Medical School at Houston guidelines. C57BL/6 mice were purchased from Jackson Lab. $Ent1^{-/-}$, $Ent2^{-/-}$, $Ent1^{flox/flox}$ and $Ent2^{flox/flox}$ were obtained from Dr. Eltzschig's lab, University of Colorado. $Ent1^{flox/flox}EpoRCre^+$ and $Ent2^{flox/flox}EpoRCre^+$ mice were generated by cross $Ent1^{flox/flox}$, $Ent2^{flox/flox}$ mice with EpoRCre-GFP mice[28] and the genotype of these are $Ent1^{flox/flox+/+}EpoRCre^+$ and $Ent2^{flox/flox+/+}EpoRCre^+$. $Adora1^{-/-}$, $Adora2a^{-/-}$, $Adora2b^{-/-}$ and $Adora3^{-/-}$ have been previously described[20]. Eight to 12 weeks, age- and sex-matched (including both male and female) C57BL/6 and mutant mice were used for experiments.

**Reagents.** NECA (catalogue #: 1691), Bay60-6583 (catalogue #: 4472) and 2′-deoxycoformycin (DCF; catalogue #: 2033) were from TOCRIS Bioscience, Bristol, UK. Dipyridamole (catalogue #: D9766), adenosine (catalogue #: A9251) and biotin NHS (catalogue #: H1759) were from Sigma-Aldrich, St. Louis, MO, USA. Forskolin (catalogue #: 3828) and dihydrochloride (H-89, catalogue #: 9844) were from Cell Signalling, Danvers, MA, USA. MG132 (catalogue #: sc-201270) and Bortezomib (catalogue #: sc-217785) were from Santa Cruz Biotechnology, Dallas, TX, USA. C14-adenosine (Moravek Biochemicals, Inc. Brea, CA; #MC-1499; specific activity: 280 mCi mmol$^{-1}$) and Percoll (GE Healthcare Life Sciences, catalog#: 17-0891-01, Piscataway, NJ, USA). ENT1 antibodies were from Santa Cruz Biotechnology (catalogue #: sc-377283. 200 ng ml$^{-1}$ of antibody was used for western blot, and 1 µg ml$^{-1}$ of antibody was used for IP) and Abcam (catalogue #: ab135756, Cambridge, MA, USA; 250 ng ml$^{-1}$ of antibody was used for western blot). ENT2 antibody was from Abcam (catalogue #: ab48595). Phospho-PKA substrate (RRXS*/T*) antibody used for IP and western blot was from Cell Signalling (catalogue #: 9624, the antibody was diluted at 1:1,000 in LI-COR blocking buffer with 0.1% Tween 20 for western blot). Ubiquitin antibodies used for western blot, IP and IF were from Santa Cruz Biotechnology (catalogue #: sc-9133 and sc-8017, for western blot: 200 ng ml$^{-1}$; for IP: 1 µg ml$^{-1}$; for IF: 1 µg ml$^{-1}$). HypoxiaProbe antibody used for IF was from HypoxiaProbe (catalogue #: 2627, 1:100 ratio of rabbit antisera was used for IF).

**Plasma adenosine extraction and measurement.** Plasma sampling, adenosine extraction and adenosine measurement were performed as described previously[20,52].

For human plasma sampling, whole blood samples were collected following 30 min of rest from a catheter placed in an antecubital vein. Blood samples were spun for 20 min at 800 $g$ at room temperature. After separation, plasma samples were stored on ice for 10 min and then stored at $-80\,^{\circ}$C for further analysis[52].

For mouse plasma sampling, mice were anesthetized with 2.5% avertin and blood was collected, in a collection tube with EDTA anticoagulant containing 1 µl of 10 mmol l$^{-1}$ dipyridamole and 1 µl of 10 mmol l$^{-1}$ ADA inhibitor deoxycoformycin (DCF), centrifuged at 2,400 $g$ for 5 min to isolate plasma, plasma samples were stored at $-80\,^{\circ}$C before adenosine extraction and analysis.

For plasma adenosine extraction, 500 µl of plasma was added to 500 µl of 0.6 mol l$^{-1}$ cold perchloric acid on ice and vortexed. The homogenate was centrifuged at 20,000 $g$ for 10 min at 4 $^{\circ}$C. The supernatant (568 µl) was transferred to a new tube and neutralized with 40.9 µl of 3 mol l$^{-1}$ KHCO$_3$ in 3.6 N KOH. Phenol red (2 µl of 0.2 mg ml$^{-1}$) was added as pH indicator. The sample was acidified with 5.7 µl of 1.8 mol l$^{-1}$ ammonium dihydrogen phosphate (pH 5.1) and 13.2 µl phosphoric acid (30%). Finally, the sample was centrifuged at 20,000 $g$ for 5 min and the supernatant was transferred to a new tube and stored at $-20\,^{\circ}$C. Before high-performance liquid chromatography (HPLC) assay, the sample was thawed on ice, and then centrifuged at 20,000 $g$ for 10 min. The supernatant was transferred to a new tube for HPLC analysis.

For plasma adenosine measurement, plasma concentration was measured using HPLC. Two hundred microlitre of adenosine sample was loaded in the HPLC and the flow rate was set at 1.5 ml min$^{-1}$. The representative peaks were identified and quantitated by running external adenosine standard curves[53].

**In vivo adenosine uptake assay.** The following protocol was approved by the Animal Welfare Committee of the University of Texas, Medical School at Houston. Mice (10–12 weeks old) were anesthetized and 100 µl C$^{14}$-Adenosine master mix (20 µmol l$^{-1}$ adenosine, 20 µCi C$^{14}$-adenosine and 200 µmol l$^{-1}$ DCF, in 0.9% saline) was injected through tail vein to achieve a final concentration of $\sim$1 µmol l$^{-1}$ adenosine, 0.1 µCi C$^{14}$-adenosine, and 10 µmol l$^{-1}$ DCF in the circulation system. After injection, 60 µl of blood was withdrawn through the right ventricle at different time points and immediately transferred to a 1.5-ml Eppendorf tube with heparin anticoagulant containing 1 µl of stop cocktail (10 mmol l$^{-1}$ DCF and 10 mmol l$^{-1}$ dipyridamole), stored on ice. Twenty microlitre of whole blood was taken and lysed in 60 µl water and the lysate was spread on a glass microfiber filter (GE Healthcare Life Sciences, catalogue number: 1825-025), heat dried for counting of C$^{14}$ isotope using a scintillation counter (LKB WALLAC 1209 EACKBETA Liquid Scintillation Counter, LKB Instruments, Victoria, Australia). The rest of the blood was centrifuged at 2,400 $g$ for 5 min at 4 $^{\circ}$C. Twenty microlitre of plasma was spread on a glass microfiber filter for C$^{14}$ isotope counting. After all plasma was removed, 20 µl of RBC was transferred into a new 1.5-ml Eppendorf tube with 80 µl 0.9% saline with 10 µmol l$^{-1}$ dipyridamole and 10 µmol l$^{-1}$ DCF; then the mixture was loaded on the top of 70 over 75% Percoll gradient, centrifuged at 2,400 $g$ for 5 min at 4 $^{\circ}$C, the supernatant was then carefully and sequentially removed. Finally, the RBCs were lysed with 60 µl water for C$^{14}$ isotope counting as mentioned previously.

**In vitro adenosine uptake assay using RBCs.** Blood was collected with heparin as an anticoagulant and centrifuged at 2,400 $g$ for 5 min at room temperature, followed by aspiration of plasma and buffy coat. Packed RBCs were then washed three times with pre-wormed culture medium at 25 $^{\circ}$C and resuspended to 4% haematocrit using same medium. One mililitre of RBC suspension was added to the wells of 12-well plates and cultured for 1 h at 25 $^{\circ}$C. The uptake assay started with transferring 54 µl of RBC suspension to an Eppendorf tube with 6 µl C$^{14}$-adenosine master mix (10 µmol l$^{-1}$ adenosine and 1 µCi ml$^{-1}$ C$^{14}$-adenosine in PBS) to geat a final adenosine concentration of 1 µmol l$^{-1}$. The uptake was performed for 30 s and stopped by adding 100 µl cold stop solution (0.9% saline with 10 µmol l$^{-1}$ dipyridamole and 10 µmol l$^{-1}$ DCF). The stopped reaction was loaded on the top of 70 over 75% percoll gradients, centrifuged at 2,400 $g$ for 5 min. The supernatant was withdrawn and the RBC pellet was processed for scintillation counting as mentioned above. Also, 54 µl of RBC suspension (washed and resuspended to 4% haematocrit as mentioned above) was aliquoted for total protein measurement using Pierce BCA Protein Assay kit (Thermo Scientific, catalogue #: 23225, Rockford, IL, USA).

**RBC membrane protein extraction and western blot.** Blood was collected with EDTA as anticoagulant. Two hundred microlitre of blood was loaded on the top of 2 ml cold 70% over 75% percoll gradient and centrifuged at 2,400 $g$ for 5 min at 4 $^{\circ}$C. The RBC pellet was washed three times with cold PBS, lysed with 540 µl cold H$_2$O with proteinase and phosphatase inhibitors, and incubated on ice for 10 min with occasional vortexing. A volume of 60 µl of 10 × PBS was added and the rest of lysate was spun at 20,000 $g$ for 20 min at 4 $^{\circ}$C. The pellet was washed three times with cold PBS and dissolved in 200 µl RIPA buffer with proteinase inhibitors (cOmplete Protease inhibitor cocktail tablets, Roche, catalogue #: 11697498001, Mannheim, Germany) and phosphatase (PhosSTOP Phosphatase inhibitor cocktail

tablets, Roche, catalogue #: 04 906 837 001, Mannheim, Germany) inhibitors, and stored at $-80\,^{\circ}$C for further analysis. Protein concentration was determined using the Pierce BCA Protein Assay kit. Protein samples were boiled in Laemml Sample Buffer (Bio-Rad, catalogue #: 161-0737, Hercules, CA) for 10 min at 95 $^{\circ}$C, separated using 10% home-made SDS–PAGE gel. Then, proteins were transferred to a nitrocellulose membrane. Western blot was performed and scanned using Odyssey Imaging System (LI-COR, Lincoln, NE) following the guidelines from the manufacture. All primary antibodies were used at 1:1,000 ratio or as indicated in Odyssey Blocking Buffer (LI-COR, catalogue #: 927-40000) with 0.2% Tween 20 and incubated at 4 $^{\circ}$C overnight. The secondary antibodies were obtained from LI-COR (IRDye 800CW Donkey anti-Rabbit IgG (H + L), catalogue #: 926-32213; IRDye 680RD Donkey anti-Mouse IgG (H + L), catalogue #: 926-68072 and all secondary antibodies were used at 100 ng ml$^{-1}$ in LI-COR blocking buffer supplied with 0.2% Tween 20.

**In vivo hypoxic challenge.** *Hypoxia treatment.* Hypoxic challenge was performed using hypoxia chamber (BioSpherix) with continued monitoring of oxygen concentration (BioSpherix, ProIx-110). Oxygen concentration was 8% for all hypoxic challenge that are $\sim$7,500 m altitude equivalent (normobaric hypoxia).

**RBC IF.** Ten microlitre fresh blood was fixed in 1 ml cold methanol for 15 min on ice. Cells were then washed two times with cold PBS and stored in PBS at 4 $^{\circ}$C. Fixed cells were blocked in 3% bovine serum albumin (BSA) in PBS for 1 h at room temperature and then incubated in primary antibody (anti-ENT1, Santa Cruz Biotechnology, 1 µg ml$^{-1}$ in 3% BSA-PBS) at 4 $^{\circ}$C overnight. After washing three times with PBS, cells were incubated in secondary antibody (Alexa 594 Donkey anti-mouse, 2 µg ml$^{-1}$ in BSA-PBS) for 1 h at room temperature. After washing, cells were spread on slides and counterstained with ProLong Gold Antifade Reagent with 4,6-diamidino-2-phenylindole (Cell Signalling, catalogue #: 8961). Images were acquired using a × 63 oil immersion objective (numerical aperture 1.4) of a Leica TCS SP5 confocal microscope (Leica Microsystems, Wetzlar, Germany).

**Assess tissue hypoxia status by hypoxia probe.** To assess tissue hypoxia status, hypoxia probe-1 (Hypoxiaprobe, Inc, Burlington, MA) 50 mg kg$^{-1}$ weight was injected to mice at 30 min before euthanizing. Tissues were isolated and fixed in 10% paraformaldehyde in PBS for at least 24 h. Fixed tissues were rinsed in PBS, dehydrated through graded ethanol washes and embedded in paraffin. Sections (5 µm) were rehydrated, permeabilized with 0.5% Triton x-100 in PBS for 20 min, blocked in 3% BSA in PBS and incubated in anti-hypoxia probe antibody (raised in rabbit, Hypoxiaprobe, Inc, Burlington, MA, 1:100 of 2,627 rabbit antisera in 3% BSA-PBS) at 4 $^{\circ}$C overnight. Goat anti-rabbit IgG (H + L) Alexa Fluor 594-conjugate antibody was used as a secondary antibody at 2 µg ml$^{-1}$ for 1 h at room temperature. After washing, the slide was counterstained with ProLong Gold Antifade Reagent with 4,6-diamidino-2-phenylindole.

**BALF collection and analysis.** For BALF collection, before sample collection, mice were anesthetized with 2.5% avertin and blood was collected. Then, lungs were lavaged four times with 0.3 ml PBS, $\sim$1 ml fluid was collected. The BALF was centrifuged and the supernatants were stored at $-80\,^{\circ}$C for further analyses[53].

For BALF total cell number counting, after BALF was collected, 40 µl BALF was aliquoted for total cell number counting using a hemacytometer under light microscopy.

BALF albumin was measured using Albuwell M kit (Exocell, Philadelphia, PA, USA) under the protocol suggested by the manufacturer.

For BALF interleukin (IL)-6 measurement, BALF IL-6 measurement was performed using Mouse IL-6 ELISA Set (BD Biosciences, San Diego, CA, USA) and according to the manufacturer's instructions. In brief, the plates were coated overnight with the 100 µl capture antibody diluted in coating buffer at 4 $^{\circ}$C and washed three times with 300 µl washing buffer. The wells then were blocked with 200 µl assay diluent for 1 h at room temperature. After the plates were washed three times, 100 µl of BALF sample or standard was added and incubated for 2 h at room temperature. Then, the plates were aspirated and washed five times. One hundred microlitre working detector (detection antibody and Streptavidin-HRP) was added to each well and incubated 1 h at room temperature, and then washed another seven times. A volume of 100 µl of substrate solution was added and incubated for 30 min at room temperature in the dark. At last, 50 µl of stop solution was added and optical density was read at 450 nm within 30 min. IL-6 concentration was calculated based on a standard curve.

**Lung MPO assay.** Pulmonary neutrophil sequestration was quantified using MPO assay as described previously[54]. In brief, animals were killed and lungs were perfused with 10 ml of PBS through the right ventricle. Lungs were snap-frozen in liquid nitrogen and stored at $-80\,^{\circ}$C. Lung was homogenized in MPO assay buffer and spun at 10,000 $g$ for 10 min at 4 $^{\circ}$C. The protein concentration of the supernatant was determined using the BAC assay kit and the supernatant was used for MPO assay using the Myeloperoxidase Colorimetric Activity Assay kit (Sigma-Aldrich, #MAK068) following the manufacturer's directions.

**Lung histology and quantification of perivascular oedema.** Mice were killed. Lung tissue was isolated and fixed in 10% paraformaldehyde in PBS for at least 24 h. Fixed lung tissue was rinsed in PBS, dehydrated through graded ethanol washes and embedded in paraffin. Paraffin-embedded lung was sectioned (5 μm). Sections were rehydrated and stained with Hematoxylin and Eosin (H&E staining).

For the quantification of perivascular oedema, the quantification was performed as described previously[55]. Perivascular oedema was defined as the fluid accumulation in the perivascular space (Fig. 2j, arrows). For the quantification of perivascular oedema, five to eight pictures of arteries from each lung section of left caudal, right caudal and right median lobes were captured at × 10 magnification on H&E-stained slides. For each perivascular oedema, the distance between adventitia and external elastic lamina was determined by measuring and averaging four measurements along the cross lines through the centre of the oedema. Then, the distance in arbitrary unit was converted to μm according to the scale bar.

**Plasma CD73 enzyme activity measurement.** CD73 enzyme activity was measured by quantifying the conversion of etheno-AMP (E-AMP) to ethenoadenosine (E-ADO) as described previously[56]. In brief, 20 μl plasma sample (collection method is described above) was incubated at room temperature with 200 nM DCF (an inhibitor of ADA) in buffer (0.1 M HEPES (pH 7.4) and 50 μM MgCl) with or without α,β-methylene ADP (APCP, Sigma-Aldrich, a specific inhibitor of CD73). Next, reactions were incubated at 37 °C for 30 min in the presence of 100 μM E-AMP. Plasma CD73 activity was determined by the changes of E-ADO concentration between reactions with or without α,β-methylene ADP. E-ADO concentration was measurement using reversed phase HPLC as describe before[57,58].

**IP–western blot.** RBCs were lysed in $H_2O$ at 1:10 volume ratio in the presence of protease inhibitor cocktail and proteasome inhibitor (N-ethylmaleimide, at 20 mM), then 10 × PBS was added to the lysate to neutralize buffer. IP–western blot was performed using 2 mg total RBC lysate in 0.5 ml IP solution (lysate and IP buffer, 50 mM (Tris-HCl, pH 7.5), 150 mM NaCl and 1% triton x-100) along with proteinase inhibitor and 20 mM N-ethylmaleimide. Lysate was pre-cleaned with 50 μl Pierce Protein A/G Agarose beads (catalogue #: 20421, ThermoFisher Scientific) at room temperature for 2 h. Ubiquitin (mouse raised, Santa Cruz, catalogue #: sc-8017, 1 μg ml$^{-1}$ of antibody was used for IP) and phospho-PKA substrate (rabbit raised, Cell Signaling, catalogue #: 9624S, antibody was used at 1:200 ratio), incubation was performed at 4 °C overnight. After primary antibody incubation, 50 μl Protein G Sepharose beads (catalogue #: 17-0618-01, GE Healthcare Life Sciences, Pittsburgh, PA, USA) was added and incubated at room temperature for 1 h. IP proteins were eluted by boiled in 30 μl 2 × Laemmli Sample Buffer (catalogue #: 1610737, Bio-Rad) for 10 min and resolved on 10% SDS–PAGE gel for western blot.

***In vivo* RBC biotin NHS labelling.** RBCs were labelled *in vivo* using NHS biotin. Specifically, 50 mg Kg$^{-1}$ of NHS biotin was injected into the retroorbital plexus of animals (prepared in 100 μl sterile saline just before injection; initially dissolved at 50 mg ml$^{-1}$ in N,-N,-dimethylacetamide) Blood samples (5 μl) were collected after first day of biotin injection from tail vein by venipuncture to determine the percentage of RBCs labelled with biotin. The percentage of biotinylated RBCs was calculated by determining the fraction of peripheral blood cells labelled with Ter-119 (to identify RBCs) that were also labelled with a streptavidin-conjugated fluorochrome by flow cytometry.

**Acclimatization hypoxic adenosine response experiment.** As illustrated in Fig. 6a, 24 h after NHS biotin injection, mice were treated with hypoxia (8% $O_2$) for 72 h, then were kept at normoxia (21% $O_2$) for either 3 days or 50 days and were given hypoxia (8% $O_2$) treatment again for 24 h. After treatment, blood samples were collected for flow cytometry, measurements of eENT1 activity, IF and plasma adenosine analysis as described previously.

**Statistical analysis.** For the human high-altitude study, we evaluated the relationship between the increase in these two biochemical parameters (that is, plasma adenosine level and sCD73 activity) to the improvement of the physiological acclimatization on Post7/21 (day 1 upon re-ascent) relative to ALT1 (day 1 of the initial ascent). Specifically, we performed Pearson correlation analysis to determine whether the elevation of plasma adenosine and sCD73 on Post7/21 relative to ALT1 was associated with the reduction of AMS-C composite score, which is commonly used to evaluate physiological adaptation to high altitude. Then, we used ordered logistic regression to further assess which specific physiological outcomes measured in the AMS-C composite score were significantly associated with plasma adenosine levels on ALT1 and Post7/21, respectively, including LLQ-Fatigue, LLQ-Headache, LLQ-Gastrointestinal, LLQ-Dizzy and LLQ-AMS Score. Finally, we applied multiple test corrections to control the FDR, and considered FDR < 0.05 as significant. All other data were presented as mean ± s.e.m. and analyzed statistically using GraphPad Prism 5 software (GraphPad Software). The significance of differences among two groups was assessed using two-tailed Student's *t*-test. Differences between the means of multiple groups were compared by one-way analysis of variance, followed by a Turkey's multiple comparisons test. Comparison of the data obtained at different time points was analyzed by two-way analysis of variance repeated measurements, followed by the Bonferroni *post hoc* test. A *P* value of < 0.05 was considered significant. Error bars in graphs are presented as mean ± s.d., unless otherwise indicated.

**Data availability.** All relevant data presented are available from the authors.

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

## Acknowledgements

This work was supported by National Institute of Health Grants HL119549 (to Y.X.), DK083559 (to Y.X.) and HL113574 (to Y.X.). Funding for the overall AltitudeOmics study was provided, in part, by grants from the U.S. Department of Defense (W81XWH-11-2-0040 Telemedicine & Advanced Technology Research Center (TATRC) to R.C.R. and W81XWH-10-2-0114 to A.T.L.); Cardiopulmonary & Respiratory Physiology Laboratory, University of Oregon; and the Charles S. Houston Endowed Professorship at the Altitude Research Center, School of Medicine, University of Colorado. L.H. was supported by Cancer Prevention & Research Institute of Texas RR150085.

## Author contributions

A.S. designed and conducted the mouse hypoxia treatment experiments, *in vitro* erythrocyte treatment experiments, western blot and IP and function of ENT measurement and analyzed all the experimental data, drew the figures and wrote the manuscript; G.G.Y. measured soluble CD3 activity in humans; H.L. measured sCD73 activity in mice. Y.Z. helped with biotin NHS injection and *in vivo* mouse hypoxia treatment; K.S. measured plasma ADA activity and helped with *in vivo* mouse hypoxia treatment; A.D. helped with adenosine determinations; J.L. helped to generate *Ent1flox/floxEpoRCre-GFP* mouse; H.K.-Q. and T.W. helped with BALF collection and related assay; T.I. helped with H.K.E. staining and tissue processing; S.Z. measured BALF IL-6 concentration; W.W. and H.W. helped with *in vivo* animal experiments, tissue collection and processing; T.N., A.W.S., S.J.-V.H., C.G.J. and A.T.L. helped with experiment design and blood sample collection in human high-altitude study; K.C.H. helped with adenosine determinations; M.B. and W.D. helped with *in vitro* C14-adenosine uptake assay and experiment design; L.H. performed statistical data analysis; J.J. helped with ubiquitin- and proteasome-related experiment design; H.Z. helped with IF and histological analysis; R.E.K. provided expertise in adenosine signalling and manuscript proofreading; H.K.E. and M.B. provided expertise in adenosine signalling and lung function and histological studies; R.C.R. led human high-altitude studies; Y.X. oversaw the design of experiments and interpretation of results, the writing and the editing of the manuscript.

## Additional information

**Competing financial interests:** The authors declare no competing financial interests.

