## [Peer Review File · Nature Communications]

Reviewers' comments:

Reviewer #1 (Remarks to the Author):

General

This manuscript reports the results of experiments aiming at understanding the phenomenon of retention of acclimatization to altitude up to some time when spending a period at low altitude. The authors have accumulated evidence in human and animal, in vivo and in vitro set-ups, that point at a mechanism allowing higher circulating adenosine levels upon re-exposure to altitude, protective against acute mountain sickness. Overall the argument is rather compelling and straightforward.

Major

The hypothesis defended by the authors is that erythrocyte purinergic memory allows to better cope with a second episode of hypoxia. It would have taken one more step to assure the alleged mechanism, namely the transfusion of blood collected at altitude / hypoxia in humans / animals after sufficient time for complete renewal of erythrocytes. Why did the authors not do this, at least in the animals (I understand that it was not done in the humans, because the mechanism was unknown at the time of the experiments).

It is unclear why only headache and dizziness were looked at and not the compound Lake Louise scores. I presume this did not come out significantly. The danger with such approaches is that by just repeating sufficiently times the testing of some sub-scores one will of course find the significance looked for.

Minor

No page numbers, no line numbers.

There are numerous typos (e.g. ubiquitination, micromolar) and problems with the English. The abstract contains redundant information.

In many places it is said that the identified mechanism can 'offset hypoxia'. That is imprecise. The hypoxia is not offset, but its consequences are.

The Introduction is imprecise in the description of the effects of altitude. For example the sentence « The inability to adjust and even death. » In the following sentence it would be good to mention « for some time » with regard to the retention of acclimatization.

The last part of the Introduction is a repetition of the well-known but so cliché mantra that this work will lead to major break-through; perhaps tone down a bit, not necessary to be so grand.

On the second page of the results section it would be good to introduce a minimum of why and how cells release ATP upon exposure to hypoxia.

When introducing the primary cultured erythrocytes mention that the cells used were already without nucleus.

It is unclear to me why this mechanism is of use for athletes engaging in strenuous exercise for gold medals, unless the exercise is at high altitude.

Reviewer #2 (Remarks to the Author):

NCOMMS-16-14844-T

Erythrocytes retain "hypoxic purinergic memory" for faster acclimatization upon re-ascent by Yang Xia and colleagues

A. Summary of the key results:

Yang Xia and colleagues have prepared a very interesting manuscript, addressing hypoxia adaptation at high altitude. Repeat journeys to high elevation are positively associated with quicker

acclimatization upon re-ascent. They propose that these protective mechanisms implicate purinergic signaling and involve mainly adenosinergic effects. These phenomena require adenosine generation from CD73 and activation of A2B receptors that impact levels of equilibrative nucleoside transporter 1 (eENT1).

Genetic deletion of eENT1 is protective via allowing the accumulation of plasma adenosine. Elegant mechanistic studies show that plasma adenosine signaling occurs via erythrocyte specific ADORA2B. This in turn induces PKA phosphorylation in red cells, ubiquitination of ENT1/several proteins and this leads to proteasomal degradation of ENT1. As erythrocytes do not have a nucleus, the levels of ENT1 remain low, which assists responses to re-ascent in humans or after re-exposure to hypoxia in mice. The authors propose that changes in eENT1 resulting from the initial hypoxia establish erythrocyte "hypoxic purinergic memory" to promote quicker and higher accumulation of plasma adenosine and faster acclimatization upon re-ascent. In essence, this work establishes a pathway of proteasomal-mediated degradation of ENT1 on erythrocytes, in the setting of hypoxia-induced adenosinergic responses and that erythrocytes retain a "hypoxic purinergic memory" for quicker adaptation with re exposure to hypoxia.

B. Originality and interest: if not novel, please give references

The authors show plasma adenosine levels are induced by high altitude and are retained at higher levels upon re-ascent in healthy individuals. They note that the erythrocyte equilibrative nucleoside transporter 1 (eENT1) protein level is decreased in both humans at high altitude and mice under hypoxic conditions. This is highly novel work, as there are no prior studies of ENT1 in altitude sickness.

ENT1 are however known to provide liver protection from ischemia and reperfusion injury. Prior work indicates ENT inhibitors may be of benefit in the prevention or treatment of ischemic liver injury, and other insults - PMID: 23703920. Cross talk also established and known to occur between ENT and A2BR - PMID: 23603835.

This work builds on the known paradigm that HIF-1-dependent repression of equilibrative nucleoside transporter (ENT) occurs in nucleated cells during hypoxia - PMID:16330813.

Authors have also shown that there are distinct relationships between adenosine signaling, oxygen dissociation and 2,3-diphosphoglycerate (2,3-DPG) in the red cell, which has been studied in subjects moving from low to high altitude and vice versa. This work is not included in the current submission. See abstract in Blood 2014 124:2664.

C. Data & methodology: validity of approach, quality of data, quality of presentation

This interesting work from an excellent group addresses altitude sickness in both man and mouse. It provides new avenues for prevention and treatment of this important condition. Studies are also pertinent for responses to hypoxia during shock and with respiratory failure.

Blood is taken with inhibitors of adenosine deaminase (deoxycoformycin) and adenosine transporters (dipyridamole). It might be of interest to present plasma levels of ATP and ADP, as well as AMP, to further validate increases in adenosine generated through CD73 actions?

In Fig. 1 we note changes in plasma adenosine levels with hypoxemia - it would be expected that during return to sea levels, that adenosine levels would remain elevated and fall off gradually during the 7/21 period if the ENT-1 was depressed in circulating red blood cells. Does this occur?

In subsequent figures, the mouse studies involve exposure of wild type (WT) and AdoraA2B or CD73-deficient mice (Cd73^{-/-}) and the targeted red cell ENT-1 nulls to hypoxia (8(-10) % oxygen, similar to that at high altitude). In the WT mice - there are substantial changes in CD73 activity, plasma adenosine levels and ENT-1 induction. These are really elegant and innovative mouse experiments, which involve erythrocyte lineage specific ablation of ENT-1; and also show that loss of CD73 or Adora2B abrogates effect of hypoxia on ENT-1.

These studies with mutant mice implicate red cell Adora2B Gs-PKA--ubiquitin-proteosomal degradation of ENT-1 (Fig. 4-6) in mediating changes in adenosine plasma fluxes and hypoxia-protection, albeit the final mechanisms of tissue protection are not dissected out in this context.

Data are of high quality and presentation is superlative

D. Appropriate use of statistics and treatment of uncertainties

Appears to be sound from my review and power analyses.

E. Conclusions: robustness, validity, reliability

The conclusions are sound and based on robust, comprehensive data.

F. Suggested improvements: experiments, data for possible revision

In line with purine-mediated beneficial effects, extracellular adenosine is also known to increase under hypoxic conditions and has an important salutary and modulatory role in hypoxia tolerance - as with preconditioning of vascularized organs. Prior work has shown that elevations in adenosine signaling contribute to the pathogenesis of preeclampsia and to sickling crises - as in PMID:25538227 and PMID: 21170046.

Why the induction of the adenosinergic pathway is injurious in sickle cell disease or eclampsia and beneficial in high altitude adaptation/preconditioning should be further clarified.

These differences may reflect that adenosine signaling might predominantly occur through A2ARs vs. A2BRs given differing levels of adenosine and disease state or pathophysiological stressors.

Also, in other associated work presented at ASH, this adenosinergic mechanism modulates 2,3-BPG levels to impact Hb-oxygen dissociation via the Rapoport-Luebering shunt. These issues as well as impacts on AMPK/HIF1alpha might be addressed a little further to develop clarity.

G. References: appropriate credit to previous work?

The bibliography is well done, is comprehensive listing prior work in this domain and due credit is given to other groups in this competitive area.

H. Clarity and context: lucidity of abstract/summary, appropriateness of abstract, introduction and conclusions.

All appear to be sound and lucid from my review.

I would just query the use of term "memory" as what we have is really an alteration or induced change in expression of ENT-1 following destruction of this membrane transporter. Would the alternative term conditioning apply?

Reviewer #3 (Remarks to the Author):

General comments

The authors report a novel adaptive response to hypoxia involving coordinated regulation of CD73 and ENT1 that achieves/maintains increased plasma adenosine, with protective effects upon tissue.

Convincing data demonstrates that this response involves sCD73-initiated increase in plasma adenosine, followed by ADORA2B-mediated phosphorylation, ubiquitination, and degradation of RBC ENT1 - which leads to 'hypoxic memory' and persisting physiology in hypoxia-exposed RBCs until they are replaced. These findings are ably demonstrated in highly complimentary studies of humans participating in the AltitudeOMICS study and in a suite of transgenic mice (with simulated ascent). The findings are quite novel and highly relevant and will be of significant interest to the community.

Specific comment follows:

Specific comments for revision

1. Results. Figure 1b. There is moderate within group diversity in adenosine levels; were there within-individual relationships between adenosine response to altitude (this is somewhat addressed by 1d, but not completely)? That is - were the individuals with the highest level on original ascent also have

highest levels upon subsequent ascent? Also, were plasma adenosine levels measured at 1,500m between ascents? Were there differences in adenosine levels based upon stay at 1,500m of either 7 or 21 days?

2. Results. Figure 1c. Were the activity assay results correlated with measurement of sCD73 protein abundance? Perhaps it is possible that the activity assay is detecting activity by another enzyme or a change in sCD73 activity w/o change in sCD73 abundance, itself.

3. Results. Murine hypoxia experiments were performed with FiO₂ of 8%, which is equivalent to that at ~ 7,500m; whereas the human studies were performed at 5,250m, at which FiO₂ is ~ 10-11%. This is, in fact a meaningful difference and it should be clearly indicated that the murine experiments model ascent that is at a substantially higher altitude than the human studies. Additionally, did the mice undergo simulated ascent over time (allowing acclimatization) as would be typical for humans (and what was the acclimatization protocol for the humans - Figure 1a)? Finally, I presume the mice were exposed to normobaric hypoxia (rather than hypobaric hypoxia)? If so, please clarify and note this as another difference from the human exposures. This is relevant since the adenosine levels in the murine experiments appear ~ 2X adenosine levels in the humans at altitude.

4. Results. Figure 2e. Please provide brief explanation of HypoxiaProbe (phosphorescence quenching) and the pO₂ threshold for the signals detected. The results in this figure are presumed to result from the increase in plasma adenosine increasing oxygen delivery to tissue - it is also possible (however unlikely) that the result, instead is from the diminished uptake of adenosine by RBCs (and altering O₂ loading/export/vasoactivity by RBCs - themselves). It is also possible that this genetic manipulation altered erythropoiesis/erythroid maturation. This possibility should be noted.

5. Results. What is meant by 'primary cultured erythrocytes'? Presumably, these are simply mature erythrocytes collected from mice and separated from plasma and other cellular components?

6. Results. Analysis of PKA-mediated phosphorylation, ubiquitination and degradation of ENT1 in humans. These results are quite interesting. Presumably, due to RBC turnover, abundance of 'hypoxia imprinted' RBCs should diminish over time following descent to 1500m after Alt16 and differ between human subjects who ascended on post 7 v post 21? Are these data available. Also, is does the loss of altitude adaptation physiology in humans (after descent) follow the same time course as RBC turnover?

Response to Reviewer 1

We appreciate the reviewer's time and effort to review our manuscript, and give us the most meaningful feedback. We are pleased that the reviewer considers "*the argument is rather compelling and straightforward*". Below is our point-to-point response to the reviewer's comment.

The hypothesis defended by the authors is that erythrocyte purinergic memory allows to better cope with a second episode of hypoxia. It would have taken one more step to assure the alleged mechanism, namely the transfusion of blood collected at altitude / hypoxia in humans / animals after sufficient time for complete renewal of erythrocytes. Why did the authors not do this, at least in the animals (I understand that it was not done in the humans, because the mechanism was unknown at the time of the experiments).

Response:

*This is a great suggestion. Suggested experiment is what we planned to do at the beginning and tried to use it to prove the principle. However, we realize that the blood transfusion in normal human subjects or mice without bleeding may not be feasible. According to **American Association of Blood Banks**, the administration of a single unit of blood (450 ml) is the standard for hospitalized people who are not bleeding which is about 10% of total blood normal human (4.7 to 5.5 liter) because transfusion of too many RBCs at one time causes multiple side effects. Thus, at this level (5 to 10%) of transfused blood, it will be difficult for us to observe the significant effects of RBCs with down-regulation of eENT1 by 72h-hypoxia treatment in the transfused animals.*

If we transfuse more blood to see the effect, for example, 50% of total blood volume, we have to bleed mice first and then conduct transfusion. However, bleeding 50% of blood in mice is a quite dangerous situation which can cause hemorrhagic shock and other related complications. Thus, it is not feasible to address the reviewer's question.

Alternatively, to prove the principle, we have used genetic tool to generate erythrocyte ENT1 knockouts, which should address the reviewer's requirement. First, we demonstrated that about 80 to 90% of erythrocyte ENT1 is ablated in $Ent1^{lox/flox}$ EpoRCre mouse that is quite similar to the level of erythrocyte ENT1 in wildtype mouse after 72h-hypoxia treatment (Fig. S2a-c and Fig. 3a). More importantly, we demonstrated that hypoxia induces much faster and higher accumulation of plasma adenosine, less hypoxyprobe staining and less tissue damage in $Ent1^{lox/flox}$ EpoRCre mice compared to the EpoRCre mice (Fig. 2d-i). Thus, we provide proof-of-principle genetic evidence that the quicker and higher elevation of plasma adenosine is beneficial to counteract hypoxic tissue damage in our erythrocyte specific eENT1 ablation mice with only around 10-20% of ENT1. We hope the reviewer recognizes the limitation and challenge of conducting blood transfusion to WT mice with or without bleeding and appreciates our effort to generate erythrocyte ENT1 (eENT1) specific knockdown to address down-regulated eENT1 in hypoxia-induced adenosine in tissue damage in vivo.

It is unclear why only headache and dizziness were looked at and not the compound Lake Louise scores. I presume this did not come out significantly. The danger with such approaches is that by

just repeating sufficiently times the testing of some sub-scores one will of course find the significance looked for.

Response:

In an effort to clarify the confusion to the reviewer, we have consult Dr. Leng Han, who is a biostatistician with substantial experience and substantially revised our manuscript to clarify the confusion as below in our result part.

Our discoveries of increased plasma adenosine and sCD73 activity in response to high altitude and their rapid and even greater elevation upon re-ascent raise a possibility that these biomarkers are physiological signals that contribute to rapid physiological acclimatization upon re-ascent. To test this possibility, we evaluated the relationship between the increase in these two biochemical parameters (i.e. plasma adenosine level and sCD73 activity) to the improvement of the physiological acclimatization on Post7/21 (day 1 upon re-ascent) relative to ALT1 (day 1 of the initial ascent). Specifically, we performed Pearson Correlation analysis to determine if the elevation of plasma adenosine and sCD73 on Post7/21 relative to ALT1 were associated with the reduction of AMC-C-Composite Score, which is commonly used to evaluate physiological adaptation to high altitude. We observed a significant correlation between the increase in plasma adenosine levels and the decrease in acute mountain sickness (AMS)-C-Composite Score on Post7/21 compared to ALT1 (Pearson Correlation $r = -0.64$, $P = 0.003$, Figure 1d). However, we did not observe a significant correlation between increased sCD73 activity with decreased AMS-C-composite Score on Post7/21 compared to ALT1 (data not shown). These data support our hypothesis that the rapid raise in plasma adenosine upon reascent represents a physiological retention for rapid acclimatization upon re-ascent.

Because elevated plasma adenosine was associated with reduced AMC-C-Composite score upon re-ascent, we used ordered logistic regression to further assess which specific physiological outcomes measured in the AMC-C-Composite Score were significantly associated with plasma adenosine levels on ALT1 and Post7/21, respectively. Among all of the categorical outcomes, including LLQ-Fatigue, LLQ-Headache, LLQ-Gastrointestinal, LLQ-Dizzy, and LLQ-AMS Score in the AMC-C-Composite Score, we found that LLQ-Headache (Ordered logistic regression, $P < 2.2e-16$, $FDR < 2.2e-16$, Figure 1e) and LLQ-Dizzy (Ordered logistic regression, $P < 2.2e-16$, $FDR < 2.2e-16$, Figure 1f) were two key physiological parameters measured in AMC-C-Composite Score that significantly associated with plasma adenosine levels on ALT1 and Post7/21, respectively. Overall, our human studies revealed that increased plasma adenosine levels are correlated to initial acclimatization and that higher levels of plasma adenosine are correlated to the rapid acclimatization upon re-ascent.

Minor

No page numbers, no line numbers.

Response:

Thanks for the suggestion. We have added the page numbers.

There are numerous typos (e.g. ubiquitination, mciromolar) and problems with the English. The abstract contains redundant information.

Response:

We have corrected all typo of “ubiquitination” to “ubiquitination”.

In many places it is said that the identified mechanism can 'offset hypoxia'. That is imprecise. The hypoxia is not offset, but its consequences are.

Response:

Thanks for the great suggestion. We changed “offset hypoxia” in our manuscript to “protect against hypoxic tissue damage”.

The Introduction is imprecise in the description of the effects of altitude. For example the sentence « The inability to adjust and even death. » In the following sentence it would be good to mention «for some time » with regard to the retention of acclimatization.

Response:

We appreciate the reviewer’s suggestion. We have modified the paragraph to “The inability to adjust to high altitude may lead to pulmonary, cerebral edema, poor cardiovascular function and even death. An intriguing and consistent observation is that following descent to lower elevations, humans retain the acclimatization to high altitude and show a much faster acclimatization upon re-ascent for some time”. Please see “Introduction, paragraph 1”.

The last part of the Introduction is a repetition of the well-known but so cliché mantra that this work will lead to major break-through; perhaps tone down a bit, not necessary to be so grand.

Response:

We respect and agree with the reviewer. In response to the reviewer’s helpful suggestion, we tone down and specifically state in this paragraph as “Thus, understanding cellular and molecular mechanisms through which altitude acclimatization occurs in normal humans may lead to new insights regarding adaptation to hypoxia and identify potential targets to counteract hypoxic tissue damage. Here, we sought to determine the common molecular basis underlying initial acclimatization and subsequent retention during re-ascent”. Please see “Introduction, paragraph 2”.

On the second page of the results section it would be good to introduce a minimum of why and how cells release ATP upon exposure to hypoxia.

Response:

- 1) We took the reviewer’s suggestion. We have added a sentence and two references to the Result (Please see “Results, second paragraph”) to explain that pannexin channels are the primary channels to release intracellular ATP to extracellular compartments under stress conditions.*
- 2) It is impossible for us to measure plasma ATP levels from human high altitude study at this moment because we have used all of samples to measure plasma adenosine and sCD73 activity. Alternatively, to address the reviewer’s question, we measured plasma ATP levels in EpoRCre (control) and Ent1^{lox/lox}EpoRCre (erythrocyte ENT1 knockout) mice at normoxia or after 72h-hypoxia treatment. We observed that plasma APT levels*

were significantly increased upon hypoxia treatment in both EpoRCre (control) and Ent1^{fllox/fllox}EpoRCre mice. These new data have been added into Figure 3c.

When introducing the primary cultured erythrocytes mention that the cells used were already without nucleus.

Response:

We have clarified in the text and specifically refer to the mature and enucleated erythrocytes. Please see “Results, subtitle ‘Adenosine signaling via ADORA2B-mediated PKA activation induces eENT1 phosphorylation, ubiquitination and proteasome degradation’, paragraph 2”.

It is unclear to me why this mechanism is of use for athletes engaging in strenuous exercise for gold medals, unless the exercise is at high altitude.

Response:

- 1) When athletes engage in strenuous exercise, physiologically, they experience relative hypoxia even at sea level due to increased oxygen consumption.*
- 2) As second reviewer pointed out our work is not limited to high altitude, “Studies are also pertinent for responses to hypoxia during shock and with respiratory failure”.*
- 3) Our current study discovered the benefit of extracellular adenosine regulates eENT1 expression of erythrocytes under hypoxia. Likely, our finding could help athletes to improve their performance. We would expect our study will be used for athletes training and hypoxic damage in near future.*

Response to Reviewer 2

We really appreciate the reviewer's time and effort to review our manuscript. The feedback and suggestion are extremely helpful and meaningful. We are pleased that the reviewer concludes *“This is highly novel work, as there are no prior studies of ENT1 in altitude sickness”* and *“It provides new avenues for prevention and treatment of this important condition. Studies are also pertinent for responses to hypoxia during shock and with respiratory failure”*. Below is our point-to-point response to the reviewer's comment.

Blood is taken with inhibitors of adenosine deaminase (deoxycoformycin) and adenosine transporters (dipyridamole). It might be interest to present plasma levels of ATP and ADP, as well as AMP, to further validate increases in adenosine generated through CD73 actions?

Response:

This is really a great suggestion.

- 1) Unfortunately, we didn't measure the levels of plasma ATP in samples from human study in the first place. The leftover plasma samples from human study is not enough for any assay and has been experienced several frozen and thaw cycles that are not suitable for any assay. Also, it is not possible for us to repeat the human study.*
- 2) Alternatively, we measured the levels of plasma ATP in samples from both $Ent1^{fllox/fllox}$ EpoRCre and control (EpoRCre) mice under normoxia and 72h-hypoxia. We found that levels of plasma ATP are increased upon 72h-hypoxia treatment. We added this result to Supplemental results (new Fig. 3c) and the description in the Results, please see “Results, subtitle ‘Hypoxia induces ubiquitination and proteasome degradation of eENT1 in vivo’, paragraph 1”.*

In Fig. 1 we note changes in plasma adenosine levels with hypoxemia - it would be expected that during return to sea levels, that adenosine levels would remain elevated and fall off gradually during the 7/21 period if the ENT-1 was depressed in circulating red blood cells. Does this occur?

Response:

This is truly a great question. We raised the same concern when we received all of the samples from human study.

- 1) Unfortunately, the original human study is not designed for our current study and there is no sample collected from Post7/21 at 1500m. It is impossible for us to repeated human study at this moment. However, it will be a great design for future study.*
- 2) As the reviewer pointed out, when we performed animal study, we did collected blood samples at different time of normoxia after 72h-hypoxia treatment. In mouse study, it does occur. In our current study, we did add two time points at Post3 and Post50 at normoxia (Fig. 6); and we did observe the phenomenon that plasma adenosine maintains high at Post3-normoxia, decreases, and back to normal level at Post50-normoxia.*

In subsequent figures, the mouse studies involve exposure of wild type (WT) and AdoraA2B or CD73-deficient mice (Cd73^{-/-}) and the targeted red cell ENT-1 nulls to hypoxia (8-10) %

oxygen, similar to that at high altitude). In the WT mice - there are substantial changes in CD73 activity, plasma adenosine levels and ENT-1 induction. These are really elegant and innovative mouse experiments, which involve erythrocyte lineage specific ablation of ENT-1; and also show that loss of CD73 or Adora2B abrogates effect of hypoxia on ENT-1.

Response:

We really appreciate the reviewer's great comments.

These studies with mutant mice implicate red cell Adora2B Gs-PKA--ubiquitin-proteosomal degradation of ENT-1 (Fig. 4-6) in mediating changes in adenosine plasma fluxes and hypoxia-protection, albeit the final mechanisms of tissue protection are not dissected out in this context.

Response:

To address the reviewer's concern, we did provide "mechanisms of tissue protection" indirectly. We used genetic tools to generate erythrocyte ENT1 knockout strain, $Ent1^{fllox/fllox}EpoRCre$, that is the equivalent of wildtype mouse after 72h-hypoxia treatment (Fig. S2a-c and Fig. 3a). We treated $Ent1^{fllox/fllox}EpoRCre$ and $EpoRCre$ (control) mice with 72h-hypoxia, then we measure several parameters that related to tissue hypoxia, pulmonary leakage, inflammation (Fig. 2 e-j). We observed the protective effects of down-regulation of erythrocyte ENT1 in acute hypoxia.

Data are of high quality and presentation is superlative

D. Appropriate use of statistics and treatment of uncertainties
Appears to be sound from my review and power analyses.

E. Conclusions: robustness, validity, reliability

The conclusions are sound and based on robust, comprehensive data.

F. Suggested improvements: experiments, data for possible revision

In line with purine-mediated beneficial effects, extracellular adenosine is also known to increase under hypoxic conditions and has an important salutary and modulatory role in hypoxia tolerance - as with preconditioning of vascularized organs. Prior work has shown that elevations in adenosine signaling contribute to the pathogenesis of preeclampsia and to sickling crises - as in PMID:25538227 and PMID: 21170046.

Response:

We really appreciate reviewer's comments. We have commented these studies and added related references in the second paragraph of Discussion.

Why the induction of the adenosinergic pathway is injurious in sickle cell disease or eclampsia and beneficial in high altitude adaptation/preconditioning should be further clarified.

These differences may reflect that adenosine signaling might predominantly occur through A2ARs vs. A2BRs given differing levels of adenosine and disease state or pathophysiological stressors.

Response:

We appreciate the reviewer's comment. To response to this comment, we have added following material into the second paragraph of Discussion:

In contrast, due to mutation of β -hemoglobin in sickle cell disease (SCD) (HbS), elevated adenosine signaling via ADORA2B-induced production of 2,3-BPG in the SCD erythrocyte becomes detrimental because it triggers deoxygenated HbS, polymerization, and eventually sickling, a central pathophysiology of SCD¹. Besides SCD, numerous studies showed that sustained accumulated adenosine signaling via ADORA2B receptors contributes to pathophysiology of multiple chronic settings including chronic kidney diseases, pulmonary fibrosis, priapism, preeclampsia and chronic pain²⁻⁶. In particular, Iriyama, et al showed that chronically elevated placental adenosine signaling via ADORA2B receptor in the trophoblasts contributes to pathophysiology of preeclampsia by inducing sFlt-1, an antiangiogenic factor known involved in preeclampsia.

How to emphasize beneficial in high altitude adaptation/preconditioning in the Discussion part? Also, in other associated work presented at ASH, this adenosinergic mechanism modulates 2,3-BPG levels to impact Hb-oxygen dissociation via the Rapoport-Luebering shunt. These issues as well as impacts on AMPK/HIF1alpha might be addressed a little further to develop clarity.

Response:

This is a great suggestion. In response to this helpful suggestion, we have added following sentence to the Second Paragraph of Discussion:

More recent studies have revealed a protective role of extracellular adenosine activating AMP-mediated protein kinase through ADORA2B receptor in the normal erythrocyte to induce 2,3-bisphosphoglycerate (2,3-BPG) production and subsequently promote oxygen delivery to counteract hypoxic tissue damages⁷.

G. References: appropriate credit to previous work?

The bibliography is well done, is comprehensive listing prior work in this domain and due credit is given to other groups in this competitive area.

Response:

We agree and appreciate the reviewer's comment.

H. Clarity and context: lucidity of abstract/summary, appropriateness of abstract, introduction and conclusions.

All appear to be sound and lucid from my review.

I would just query the use of term "memory" as what we have is really an alteration or induced change in expression of ENT-1 following destruction of this membrane transporter. Would the alternative term conditioning apply?

Response:

We take the reviewer's suggestion and re-think regarding the use of word "memory". In response to the reviewer's suggestion, we change "hypoxic purinergic memory" to "hypoxic adenosine response".

Response to Reviewer #3

We are pleased that the reviewer considers “*our findings are quite novel and highly relevant and will be of significant interest to the community*”. Below is our point-to-point response to each comment.

1. Results. Figure 1b. There is moderate within group diversity in adenosine levels; were there within-individual relationships between adenosine response to altitude (this is somewhat addressed by 1d, but not completely)? That is - were the individuals with the highest level on original ascent also have highest levels upon subsequent ascent?

Response:

To address the reviewer’s concern, we used Spearman Correlation to test the associations between individuals at sea level (SL) and ALT16. We observed a significant correlation (Spearman correlation $r=0.434$, $p\text{-value}=0.038$), suggested that individuals with the high level on original ascent also have relative high levels upon subsequent ascent.

Also, were plasma adenosine levels measured at 1,500m between ascents? Were there differences in adenosine levels based upon stay at 1,500m of either 7 or 21 days?

Response:

This is the same question raised by reviewer 2. We raised the same concern when we received all samples from human study.

- 1) Unfortunately, the original human study is not designed for our current study and there is no sample collected from Post7/21 at 1500m. There is no possibility for us to repeated human study. It is impossible for us to repeated human study at this moment. However, it will be a great design for future study.*
- 2) As the reviewer pointed out, when we performed animal study, we did collected blood samples at different time of normoxia after 72h-hypoxia treatment. In mouse study, it does occur. In our current study, we did add two time points at Post3 and Post50 at*

normoxia (Fig. 6); and we did observe the phenomenon that plasma adenosine maintains high at Post3-normoxia, decreases, and back to normal level at Post50-normoxia.

2. Results. Figure 1c. Were the activity assay results correlated with measurement of sCD73 protein abundance? Perhaps it is possible that the activity assay is detecting activity by another enzyme or a change in sCD73 activity w/o change in sCD73 abundance, itself.

Response:

These are great suggestion regarding sCD73 activity.

- 1) Due to the quantity (we almost used up the plasma samples for various analysis) and quality (the leftover of plasma samples are experienced several frozen and thaw cycles) of plasma samples from human study, it is not possible for us to access the abundance and modification of sCD73 in the plasma.*
- 2) The method we employed for measuring sCD73 activity is very specific, in that we compare the amount of conversion etheno-AMP (E-AMP) to ethenoadenosine (E-ADO) with or without the present of α,β -methylene ADP (APCP, a specific inhibitor of CD73)⁸. By using this method, we can accurately measure sCD73 activity in plasma samples.*

3. Results. Murine hypoxia experiments were performed with FiO₂ of 8%, which is equivalent to that at ~ 7,500m; whereas the human studies were performed at 5,250m, at which FiO₂ is ~ 10-11%. This is, in fact a meaningful difference and it should be clearly indicated that the murine experiments model ascent that is at a substantially higher altitude than the human studies.

Response:

This is great suggestion. It is true that FiO₂ of 8% is equivalent to about 7,500m. We appreciate and take the reviewer's suggestion, modify the text in the manuscript. Specifically, we have added the following sentence to the result part "Of note, we found that plasma adenosine levels were induced to higher levels than in humans under hypoxia (Fig.1b and Fig. 2d). This difference is likely due to mice exposed to 8% normobaric hypoxia equivalent to 7500m which is higher than humans exposed to 5,260m equivalent to 10% hypobaric hypoxia". Please see "Results, subtitle 'Genetic deletion of erythrocyte ENT1 leads to rapid accumulation of extracellular adenosine under acute hypoxia and counteracts hypoxia-induced tissue damage in mice', paragraph 2". Also, in Supplementary Material, we add "that are about 7,500m altitude equivalent (normobaric hypoxia)". Please see "Material and Methods, subtitled "in vivo hypoxic challenge".

Additionally, did the mice undergo simulated ascent over time (allowing acclimatization) as would be typical for humans (and what was the acclimatization protocol for the humans - Figure 1a)?

Response:

We didn't let mice undergo simulated ascent over time (allowing acclimatization) as in human study.

Finally, I presume the mice were exposed to normobaric hypoxia (rather than hypobaric hypoxia)? If so, please clarify and note this as another difference from the human exposures. This is relevant since the adenosine levels in the murine experiments appear ~ 2X adenosine levels in the humans at altitude.

Response:

We take the reviewer's suggestion. Specifically, we have added the following sentence to the result part "Of note, we found that plasma adenosine levels were induced to higher levels than in humans under hypoxia (Fig.1b and Fig. 2d). This difference is likely due to mice exposed to 8% normobaric hypoxia equivalent to 7500m which is higher than humans exposed to 5,260m equivalent to 10% hypobaric hypoxia". Please see "Results, subtitle 'Genetic deletion of erythrocyte ENT1 leads to rapid accumulation of extracellular adenosine under acute hypoxia and counteracts hypoxia-induced tissue damage in mice', paragraph 2".

4. Results. Figure 2e. Please provide brief explanation of HypoxiaProbe (phosphorescence quenching) and the pO₂ threshold for the signals detected. The results in this figure are presumed to result from the increase in plasma adenosine increasing oxygen delivery to tissue - it is also possible (however unlikely) that the result, instead is from the diminished uptake of adenosine by RBCs (and altering O₂ loading/export/vasoactivity by RBCs - themselves).

Response:

We appreciate the reviewer's helpful comment and suggestion.

First, the principle of HypoxiaProbe is that the 2-nitroimidazoles form adducts with thiol groups in proteins, peptides and amino acids in hypoxic cells in vitro and in vivo that are ready to be detected by an antibody⁹. Hypoxia (pO₂ < 10 mmHg) is required for the binding^{10,11}.

Second, Previous study showed that extracellular adenosine through ADORA2B receptor caused 2,3-BPG induction, subsequently increased P50 (lower oxygen-hemoglobin binding affinity) and reduced tissue hypoxia.⁷ In current study, unfortunately we didn't measured P50 in our animal study; but it is likely, rapidly accumulated extracellular adenosine observed in Ent1^{fllox/fllox}EpoRCre-GFP mice under acute hypoxia would increase oxygen delivery to hypoxic tissues through adenosine-ADORA2B signaling and 2,3-BPG induction. As result showed in Fig. 2e, less HypoxiaProbe intensity was observed in Ent1^{fllox/fllox}EpoRCre-GFP mice under acute hypoxia than EpoRCre-GFP control mice.

It is also possible that this genetic manipulation altered erythropoiesis/erythroid maturation. This possibility should be noted.

Response:

This is an important question regarding molecular basis underlying erythropoiesis under hypoxia. At this moment, we have not focused on hypoxia-induced erythropoiesis. Instead, we have mainly focused on ADORA2B signaling via PKA-mediated proteasomal degradation of ENT1 in mature erythrocytes. We feel this will be quite important future question to follow-up.

5. Results. What is meant by 'primary cultured erythrocytes'? Presumably, these are simply mature erythrocytes collected from mice and separated from plasma and other cellular components?

Response:

Yes, the reviewer is right. We specifically refer that mature erythrocytes isolated from mice and separated from plasma and other blood components using 70 over 75% percoll density gradient.

6. Results. Analysis of PKA-mediated phosphorylation, ubiquitination and degradation of ENT1 in humans. These results are quite interesting. Presumably, due to RBC turnover, abundance of 'hypoxia imprinted' RBCs should diminish over time following descent to 1500m after Alt16 and differ between human subjects who ascended on post 7 v post 21? Are these data available. Also, is does the loss of altitude adaptation physiology in humans (after descent) follow the same time course as RBC turnover?

Response:

These are really great questions.

The original purpose of human study did not design for our current study and it is impossible for us to repeat the human study. Unfortunately, we can't answer your questions by using human study.

As alternatives, we followed the RBC life span in our animal study (Fig. 6). We observed the loss of altitude adaptation physiology in animals (after hypoxia and back to normoxia) when pre-exposed RBCs were replaced by newly formed RBCs after 55days (mouse RBC's life span is round 55days). Concurrently, we found that eENT1 levels and activity gradually return to normal levels as normoxia and in turn the retain a rapid elevation of adenosine was lost post 50 day first hypoxia exposure.

References

- 1 Zhang, Y. *et al.* Detrimental effects of adenosine signaling in sickle cell disease. *Nat Med* **17**, 79-86, doi:10.1038/nm.2280 (2011).
- 2 Iriyama, T. *et al.* Elevated placental adenosine signaling contributes to the pathogenesis of preeclampsia. *Circulation* **131**, 730-741, doi:10.1161/circulationaha.114.013740 (2015).
- 3 Dai, Y. *et al.* A2B adenosine receptor-mediated induction of IL-6 promotes CKD. *Journal of the American Society of Nephrology : JASN* **22**, 890-901, doi:10.1681/asn.2010080890 (2011).
- 4 Karmouty-Quintana, H. *et al.* Deletion of ADORA2B from myeloid cells dampens lung fibrosis and pulmonary hypertension. *Faseb j* **29**, 50-60, doi:10.1096/fj.14-260182 (2015).
- 5 Ning, C. *et al.* Excess adenosine A2B receptor signaling contributes to priapism through HIF-1alpha mediated reduction of PDE5 gene expression. *Faseb j* **28**, 2725-2735, doi:10.1096/fj.13-247833 (2014).
- 6 Hu, X. *et al.* Sustained Elevated Adenosine via ADORA2B Promotes Chronic Pain through Neuro-immune Interaction. *Cell reports* **16**, 106-119, doi:10.1016/j.celrep.2016.05.080 (2016).
- 7 Liu, H. *et al.* Beneficial Role of Erythrocyte Adenosine A2B Receptor-Mediated AMP-Activated Protein Kinase Activation in High-Altitude Hypoxia. *Circulation* **134**, 405-421, doi:10.1161/circulationaha.116.021311 (2016).

- 8 Zhang, W. *et al.* Elevated Ecto-5'-nucleotidase-Mediated Increased Renal Adenosine Signaling Via A2B Adenosine Receptor Contributes to Chronic Hypertension. *Circulation Research* **112**, 1466-1478, doi:10.1161/circresaha.111.300166 (2013).
- 9 Varghese, A. J., Gulyas, S. & Mohindra, J. K. Hypoxia-dependent reduction of 1-(2-nitro-1-imidazolyl)-3-methoxy-2-propanol by Chinese hamster ovary cells and KHT tumor cells in vitro and in vivo. *Cancer Res* **36**, 3761-3765 (1976).
- 10 Raleigh, J. A., Franko, A. J., Koch, C. J. & Born, J. L. Binding of misonidazole to hypoxic cells in monolayer and spheroid culture: evidence that a side-chain label is bound as efficiently as a ring label. *Br J Cancer* **51**, 229-235 (1985).
- 11 Varghese, A. J. Glutathione conjugates of misonidazole. *Biochem Biophys Res Commun* **112**, 1013-1020 (1983).

REVIEWERS' COMMENTS:

Reviewer #1 (Remarks to the Author):

The authors have satisfactorily replied to the criticisms raised.

Some minor things:

Line 44, wouldn't '...to counteract hypoxia induced maladaptation.' be more correct? At least for the human results?

Line 56, rather '...lead to pulmonary or cerebral edema ...'

Line 61, put '(CaO₂ is lower)' after 'erythropoiesis'

Line 71, delete 'to' after 'facing'

Line 77, also here I suggest to use '...to counteract the maladaptive effects of hypoxia' or so, because there is not necessarily tissue damage, even though there can be, and was in the animal data.

Line 82, 'is associated'

Line 128, should probably read 'acute mountain sickness (AMS)-C composite score' and on line line 131 'AMS-C composite score'

Line 135, 're-ascent'

Line 143, also here correct to 'AMS-C composite score'

Line 156, also here I suggest not to use the word 'damage' since you did not see any, but rather use 'maladaptation' or so, at least for the human results.

Line 173, please round this percentage

Line 210, 'about an altitude of 7'500m)

Line 215, 'to the mice being exposed to 8% normobaric hypoxia, equivalent to an altitude 7500m, which was higher than the humans who were exposed to 5,260m equivalent to 10% hypobaric hypoxia.'

Line 334, 'the mouse finding'

Line 337, 'findings'

Line 349, 'an "erythrocyte adenosine response"'

Line 350, something wrong here with the syntax

Line 352, 'an "hypoxic adenosine response "'

Line 353, 'are replaced' and 'higher levels of'

Line 354, 'mimicked' and 'by exposing WT'

Line 355, 'bringing' and 're-exposing'

Line 359, 'but not to significantly increase erythropoiesis'

Line 395, 'signals via ADORA2B, inducing '

Line 399, 'an erythrocyte'

Line 400, 'provides'

Line 402, something wrong with this sentence

Line 405, 'the effects of hypoxia.'

Line 410, 'settings'

line 474, 'erythrocyte have no nuclei'

Line 523, 'the potential consequences of exposure to hypoxia'

Reviewer #2 (Remarks to the Author):

The comments and criticisms have been addressed to my satisfaction

The work is of high quality and of great interest

Reviewer #3 (Remarks to the Author):

All my concerns have been adequately addressed. I congratulate the authors on an excellent manuscript.

Response to Reviewer 1

We appreciate the reviewer's time and effort to review our manuscript, and give us feedback. Below is our point-to-point response to the reviewer's comment. We made all changes as Reviewer 1 pointed out in the text.

Line 44, wouldn't '...to counteract hypoxia induced maladaptation.' be more correct? At least for the human results?

Response:

We made the change, changed “hypoxic tissue damage” to “hypoxia induced maladaptation”.

Line 56, rather '...lead to pulmonary or cerebral edema ...'

Response:

We made the change as the reviewer pointed out.

Line 61, put '(CaO₂ is lower)' after 'erythropoiesis'

Response:

We made the change as the reviewer pointed out.

Line 71, delete 'to' after 'facing'

Response:

We made the change and delete “to”.

Line 77, also here I suggest to use '...to counteract the maladaptive effects of hypoxia' or so, because there is not necessarily tissue damage, even though there can be, and was in the animal data.

Response:

We made changes throughout the manuscript as the reviewer pointed out.

Line 82, 'is associated'

Response:

We made the change as the reviewer pointed out.

Line 128, should probably read 'acute mountain sickness (AMS)-C composite score' and on line

Response:

We made changes throughout the manuscript as the reviewer pointed out.

line 131 'AMS-C composite score'

Response:

We made the change.

Line 135, 're-ascent'

Response:

We corrected the typo.

Line 143, also here correct to 'AMS-C composite score'

Response:

We made the change as the reviewer pointed out.

Line 156, also here I suggest not to use the word 'damage' since you did not see any, but rather use 'maladaptation' or so, at least for the human results.

Response:

We made changes throughout our manuscript as the reviewer pointed out.

Line 173, please round this percentage

Response:

We made the change and round the percentage.

Line 210, 'about an altitude of 7'500m)

Response:

We made changes as the reviewer pointed out throughout the manuscript for expression of altitude.

Line 215, 'to the mice being exposed to 8% normobaric hypoxia, equivalent to an altitude 7500m, which was higher than the humans who were exposed to 5,260m equivalent to 10% hypobaric hypoxia.'

Response:

We made the change as the reviewer indicated.

Line 334, 'the mouse finding'

Response:

We made the change.

Line 337, 'findings'

Response:

We made the change.

Line 349, 'an “erythrocyte adenosine response”'

Response:

We made the change.

Line 350, something wrong here with the syntax Line 352, 'an “hypoxic adenosine response”'

Response:

As the reviewer pointed out, we rewrote the following sentences “Thus, our human in vivo high altitude studies and mouse genetic studies raise a novel but compelling concept that 1) hypoxia-mediated degradation of eENT1 establishes “erythrocyte adenosine response” with an extensive pool of erythrocytes with reduced ability to eliminate plasma adenosine, promote rapid accumulation of plasma adenosine and in turn retain faster acclimatization upon re-ascent; 2) erythrocytes, as the most abundant cells in our body, retain “hypoxic adenosine response” for a certain time period until hypoxia-exposed RBCs replaced by newly synthesized RBCs containing high ENT1 proteins”. As “Thus, our human in vivo high altitude studies and mouse genetic studies raise a novel but compelling hypothesis that erythrocytes, as the most abundant cells in our body, retain hypoxic adenosine response for a certain time period until hypoxia-exposed RBCs are replaced by newly synthesized RBCs containing higher levels of ENT1 proteins.”

Line 353, 'are replaced' and 'higher levels of'

Response:

We made changes as the reviewer indicated.

Line 354, 'mimicked' and 'by exposing WT'

Response:

We made changes as the reviewer pointed out.

Line 355, 'bringing' and 're-exposing'

Response:

We made the change as the reviewer indicated.

Line 359, 'but not to significantly increase erythropoiesis'

Response:

We made the change as the reviewer indicated.

Line 395, 'signals via ADORA2B, inducing '

Response:

We made the change as the reviewer indicated.

Line 399, 'an erythrocyte'

Response:

We made the change as the reviewer indicated.

Line 400, 'provides'

Response:

We made the change as the reviewer indicated.

Line 402, something wrong with this sentence Line 405, 'the effects of hypoxia.'

Response:

We made the change as the reviewer indicated and rewrote this part. As “The reduced eENT1 resulting from the initial hypoxic exposure establishes an erythrocyte hypoxic adenosine

response for a 2nd hypoxic exposure, and underlies a faster and improved acclimatization upon re-ascent associated with high levels of circulating adenosine (Fig. 7).”

Line 410, 'settings'

Response:

We made the change as the reviewer indicated.

line 474, 'erythrocyte have no nuclei'

Response:

We made the change as the reviewer indicated.

Line 523, 'the potential consequences of exposure to hypoxia'

Response:

We made the change as the reviewer indicated.

Response to Reviewer 2

Reviewer #2 (Remarks to the Author):

The comments and criticisms have been addressed to my satisfaction The work is of high quality and of great interest.

Response:

We really appreciate the reviewer's time, efforts and comments.

Response to Reviewer 3

Reviewer #3 (Remarks to the Author):

All my concerns have been adequately addressed. I congratulate the authors on an excellent manuscript.

Response:

We really appreciate the reviewer's time, efforts and comments.